# Fixational eye movements enhance the precision of visual information transmitted by the primate retina

Eric G. Wu [1] ✉, Nora Brackbill [2], Colleen Rhoades [3], Alexandra Kling[4,5,6], Alex R. Gogliettino[6,7], Nishal P. Shah [1,4], Alexander Sher [8], Alan M. Litke [8], Eero P. Simoncelli [9,10,11] & E. J. Chichilnisky [4,5,6] ✉

Fixational eye movements alter the number and timing of spikes transmitted from the retina to the brain, but whether these changes enhance or degrade the retinal signal is unclear. To quantify this, we developed a Bayesian method for reconstructing natural images from the recorded spikes of hundreds of retinal ganglion cells (RGCs) in the macaque retina (male), combining a likelihood model for RGC light responses with the natural image prior implicitly embedded in an artificial neural network optimized for denoising. The method matched or surpassed the performance of previous reconstruction algorithms, and provides an interpretable framework for characterizing the retinal signal. Reconstructions were improved with artificial stimulus jitter that emulated fixational eye movements, even when the eye movement trajectory was assumed to be unknown and had to be inferred from retinal spikes. Reconstructions were degraded by small artificial perturbations of spike times, revealing more precise temporal encoding than suggested by previous studies. Finally, reconstructions were substantially degraded when derived from a model that ignored cell-to-cell interactions, indicating the importance of stimulus-evoked correlations. Thus, fixational eye movements enhance the precision of the retinal representation.

Vision begins with the retina, which transforms dynamic visual images into electrical signals, processes these signals, and transmits them to the brain in the spiking activity of retinal ganglion cells (RGCs). This encoding process has been studied for nearly a century, with contemporary models capturing the details of RGC responses with a high degree of precision. But these models do not directly reveal how effectively the visual scene is conveyed by RGCs particularly under stimulus conditions that the visual system evolved to analyze: naturally

occurring patterns of light, with dynamic global image shifts arising from eye movements.

To probe the retinal code under these conditions, we develop and apply a method for reconstructing natural images and movies from the spiking activity of complete populations of RGCs recorded in the primate retina, in response to continuously changing visual images. Rather than using a regression formalism to optimize a decoding model that maps recorded RGC spikes to images[1–3], we use a Bayesian

[1]Department of Electrical Engineering, Stanford University, Stanford, CA, USA. [2]Department of Physics, Stanford University, Stanford, CA, USA. [3]Department of Bioengineering, Stanford University, Stanford, CA, USA. [4]Department of Neurosurgery, Stanford University, Stanford, CA, USA. [5]Department of Ophthalmology, Stanford University, Stanford, CA, USA. [6]Hansen Experimental Physics Laboratory, Stanford University, 452 Lomita Mall, Stanford 94305 CA, USA. [7]Neurosciences PhD Program, Stanford University, Stanford, CA, USA. [8]Santa Cruz Institute for Particle Physics, University of California, Santa Cruz, Santa Cruz, CA, USA. [9]Flatiron Institute, Simons Foundation, New York, NY, USA. [10]Center for Neural Science, New York University, New York, NY, USA. [11]Courant Institute of Mathematical Sciences, New York University, New York, NY, USA. ✉e-mail: wu.eric.g@gmail.com; ej@stanford.edu

formalism[4]−combining a likelihood model that accounts for the RGC spikes with a separate prior model that captures the statistical structure of natural images. Specifically, images are reconstructed by numerical optimization of the posterior density, arising from the product of (1) an image likelihood obtained from an encoding model fitted to RGC data that captures the stochastic responses of RGCs to visual stimuli[5], and (2) a natural image prior implicit in an artificial neural network pre-trained on a natural image database to perform denoising[6]. This approach confers unique advantages for the analysis and interpretation of the retinal signals. We demonstrate that the method achieves state-of-the-art reconstruction performance, and then use it to quantify the importance of fixational eye movements, spike timing precision, and cell-to-cell correlations in the retinal code for natural visual stimuli.

## Results

To characterize the visual signals evoked by natural images, we recorded light responses of RGCs in isolated peripheral macaque retina with a large-scale multi-electrode array[7]. This method captured the activity of nearly complete populations of several hundred RGCs of the four numerically dominant types (ON midget, OFF midget, ON parasol, OFF parasol), which comprise roughly 70% of RGC axons projecting to the brain[8]. Spatiotemporal white noise stimuli were used to identify cells and map their receptive fields[9,10].

### Bayesian reconstruction of flashed images

We first examined the reconstruction of images presented in brief flashes to the retina. Although the dynamics of the flashed stimulus differ markedly from the natural visual experience, the simplicity of the stimulus enabled the evaluation of the image reconstruction approach and comparison to previous methods. Thousands of grayscale photographic images from the ImageNet database[11,12] were presented for a duration of 100 ms with consecutive trials separated by 400 ms of the uniform gray screen (Fig. 1a, also see the "Methods" section).

Flashed natural images were reconstructed from evoked RGC activity using a Bayesian approximate maximum *a posteriori* (MAP) algorithm (see ref. 13). The posterior density (probability of an image given observed spikes) is the product of two separately defined and estimated components: (1) a likelihood model of the natural image stimulus $\mathbf{y}$ evoking the measured spiking response $\mathbf{s}$, $p(\mathbf{s}|\mathbf{y})$, computed using a probabilistic encoding model of RGC spiking in response to natural image stimuli; (2) a prior model of natural images, $p(\mathbf{y})$, obtained implicitly from a Gaussian-denoising neural network (Fig. 1c). The likelihood was computed from an encoding model that summed the effects of the visual input, spike history, and spike trains of nearby neurons (to capture spike train temporal structure and cell-to-cell correlations) and then transformed the output with an instantaneous sigmoidal nonlinearity to provide a firing probability for a Bernoulli spike generator (Fig. 1b). This model generalizes the commonly used linear−nonlinear-Poisson (LNP) cascade model, replacing Poisson spiking with Bernoulli spiking (equivalent at fine time scales), and is a specific case of a generalized linear model (GLM), incorporating recursive feedback and coupling filters[5]. We refer to this as the linear−nonlinear-Bernoulli with recurrent coupling (LNBRC) model. Model parameters (stimulus, feedback, and coupling filters, and an additive constant) were jointly fitted to recorded RGC data by maximizing the likelihood of the model parameters given the stimulus and the observed spikes, augmented with regularization terms to induce sparsity in the filter weights (see the "Methods" section). Separately, an implicit image prior was obtained by training a denoising convolutional neural network (dCNN) to remove additive Gaussian noise from a large collection of natural images[6]. Such priors underlie the "diffusion models"[14] that represent the current state-of-the-art in machine learning for image synthesis[15,16] and inference[17–19].

With these two components, the reconstruction procedure maximized the posterior by alternating between an encoding likelihood optimization step (solved with unconstrained convex minimization) and a prior optimization step (solved with a single forward pass of the denoiser[6,20]) (Fig. 1d, e, see the "Methods" section), yielding an estimate of the most probable image given the RGC spikes and natural image statistics.

The performance of the MAP reconstruction algorithm was characterized qualitatively with visual image comparison and quantitatively with MS-SSIM[21], a commonly used measure of perceptual image quality. Example reconstructions are shown in Fig. 1f. Reconstruction performance with this MAP procedure was qualitatively and quantitatively more accurate than that obtained using linear reconstruction[1,3,4] (mean MS-SSIM of 0.685, 0.652, 0.660, and 0.652 for LNBRC-dCNN MAP reconstructions in the four experimental preparations tested, compared to 0.624, 0.616, 0.578, and 0.575 for linear reconstruction in the same preparations). Performance was comparable to state-of-the-art neural networks trained to nonlinearly recover the high spatial frequency components of images[2] (mean MS-SSIM of 0.689, 0.683, 0.651, and 0.653, in the same preparations). In addition to reconstruction quality, the MAP approach provided greater interpretability by separating the likelihood and prior components of estimation, and broader usability with limited retinal data (the retinal encoding model contained ~1.5 million parameters, in comparison with ~240 million parameters for the benchmark neural network model[2]).

To examine the importance of the encoding and prior models, MAP reconstruction performance with the full model (labeled LNBRC-dCNN) was compared to that achieved with a simpler spectral Gaussian image prior (LNBRC-1F) or with a likelihood corresponding to a simpler LNP encoding model (LNP-dCNN). Images reconstructed using the full approach had sharper and more detailed image structure (edges, contours, textures) than those reconstructed using the 1/F prior, and contained more fine spatial detail than those reconstructed using the LNP encoding model (Fig. 1f). Quantitatively, reconstructions produced using LNBRC-dCNN exhibited greater similarity to the original image than those produced with the simpler 1/F prior (mean MS-SSIM of 0.612, 0.573, 0.577, and 0.565 for LNBRC-1F in each of four experimental preparations, all *p*-values $< 1 \times 10^{-10}$, LNBRC-dCNN > LNBRC-1F, Wilcoxon signed rank test, $N = 1500$, $N = 1750$, $N = 750$, and $N = 750$, respectively) or the simpler LNP encoding model (mean MS-SSIM of 0.635, 0.613, 0.597, and 0.603 using LNP-dCNN in the same preparations, all *p*-values $< 1 \times 10^{-10}$, LNBRC-dCNN > LNP-dCNN, Wilcoxon signed rank test, $N = 1500$, $N = 1750$, $N = 750$, and $N = 750$, respectively). Thus, both the dCNN image prior and the LNBRC encoding model contribute substantially to the quality of natural image reconstruction.

### Bayesian reconstruction of images displayed with fixational eye movements

Fixational (ocular) drift, the small but incessant eye movements that occur when fixating a visual target, is a fundamental component of primate vision. Recent studies have demonstrated that these eye movements enhance fine pattern vision and visual acuity[22–26], and have hypothesized that these effects could result from sampling the image at many spatial phases relative to the lattice of RGC receptive fields[24,25,27], and/or from modulating high frequency spatial details into the temporal domain[22,23]. However, psychophysical studies[25,28–30] suggest that the visual system may not have precise knowledge of the eye position (but see ref. 31), opening the possibility that positional uncertainty could instead degrade the retinal signal[32] (but see ref. 33). Although simulation studies[27,34–36] have explored the possibility of using the retinal signal alone to compensate for fixational eye movements, it remains uncertain whether unknown eye jitter enhances or degrades the retinal representation itself. Here, we directly characterized the effects of fixational drift eye movements by

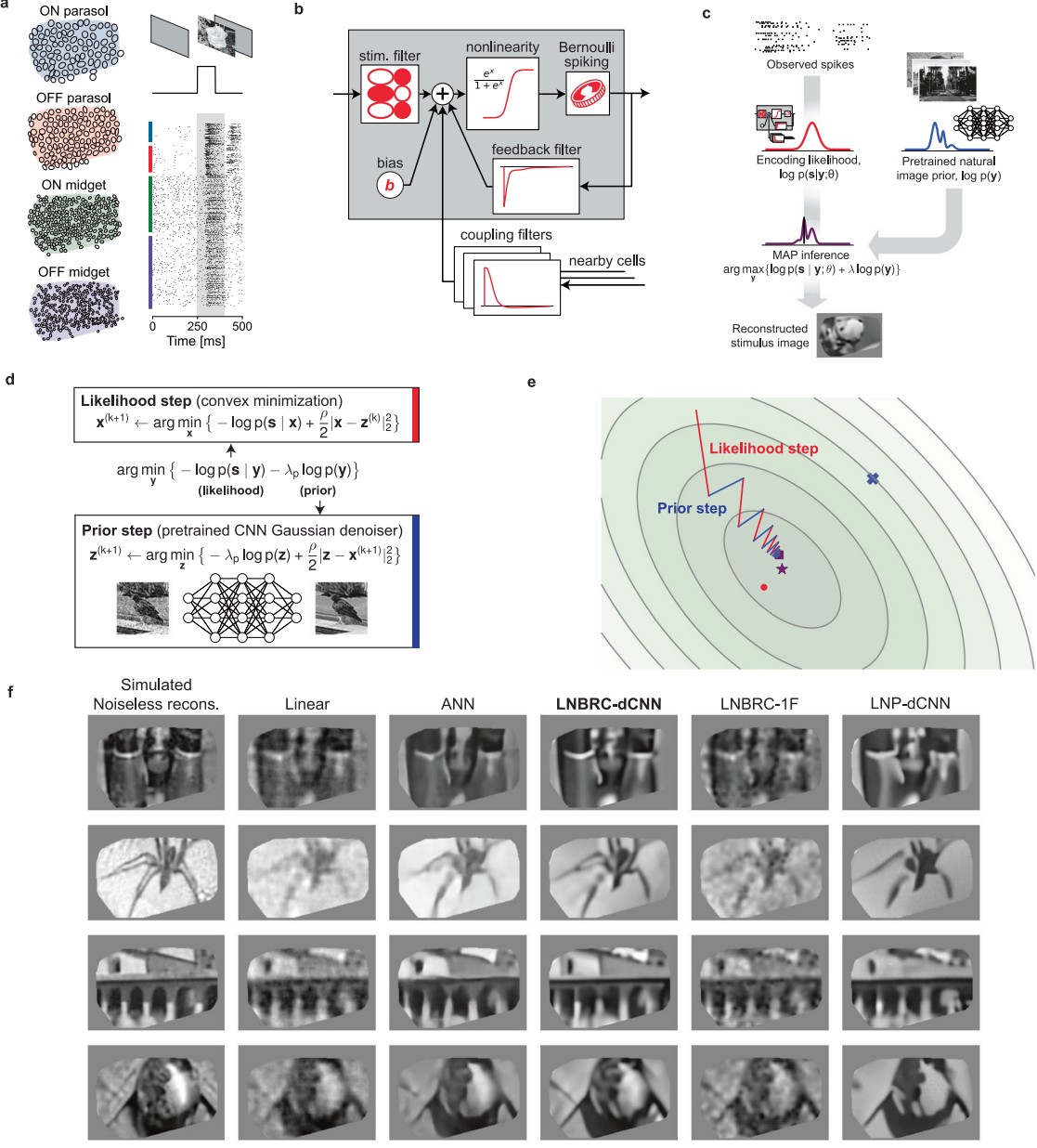

**Fig. 1 | Reconstruction of flashed natural images from RGC spikes. a** Example of macaque retinal data. Receptive field mosaics for the major RGC types (ON parasol, OFF parasol, ON midget, OFF midget). Natural images are flashed for 100 ms, and spikes recorded from all 691 cells over a 150 ms interval (gray region) were used for LNBRC model fitting and reconstruction. **b** LNBRC encoding model. Model cell responses are computed from the spatio-temporally filtered visual stimulus, combined with filtered spike trains from the cell and neighboring cells. These filtered spiking inputs capture both spike train temporal structure and cell-to-cell correlations. **c** Bayesian reconstruction. The likelihood computed using the LNBRC encoding model is combined with a separately trained natural image prior to producing a posterior density for the stimuli given observed spike trains. **d** Half-quadratic variable splitting algorithm for approximate MAP optimization. The method alternates between optimizing the likelihood (a convex minimization problem, solved using gradient descent), and optimizing the prior probability (by applying an artificial neural network pre-trained to perform Gaussian denoising on

natural images). **e** Visualization of the optimization path for a highly simplified two-dimensional toy problem (red lines are likelihood steps, blue lines are prior steps). The contours indicate level sets of the posterior, with the mode of posterior (purple star), likelihood (red dot), and prior (blue x). The step size progressively decreases, corresponding to increasing values of schedule hyperparameter ρ. **f** Example reconstructions comparing LNBRC-dCNN with benchmarks and alternative models. Columns: Simulated noiseless reconstruction, a reconstruction of the stimulus from linear projections onto the LNBRC filters (see also the "Methods" section); Linear reconstruction, a simple benchmark; ANN, direct artificial neural network reconstruction[2]; LNBRC-dCNN, our Bayesian method; LNBRC-1F, Bayesian method with the dCNN image prior with a simpler 1/F Gaussian image prior; and LNP-dCNN, replacing the LNBRC likelihood with a simpler LNP likelihood. The original stimuli were taken from the ImageNet dataset[11] and are not displayed due to copyright restrictions. The complete comparisons are available at https://github.com/wueric/wu-nature-comms-2024.

reconstructing images from the experimentally recorded responses of RGCs to jittered natural images.

RGC activity was measured in response to movies consisting of images from the ImageNet database[11,12], displayed with randomly jittered spatial offsets in each frame to emulate fixational drift eye

movements. Images were displayed for 500 ms, with each 8.33 ms frame spatially shifted relative to the previous frame according to a discretized sample from a 2D Gaussian distribution to approximate a Brownian motion with a diffusion constant of 10 μm²/frame (Fig. 2ab). This design approximates drift eye movements as a 2D Brownian

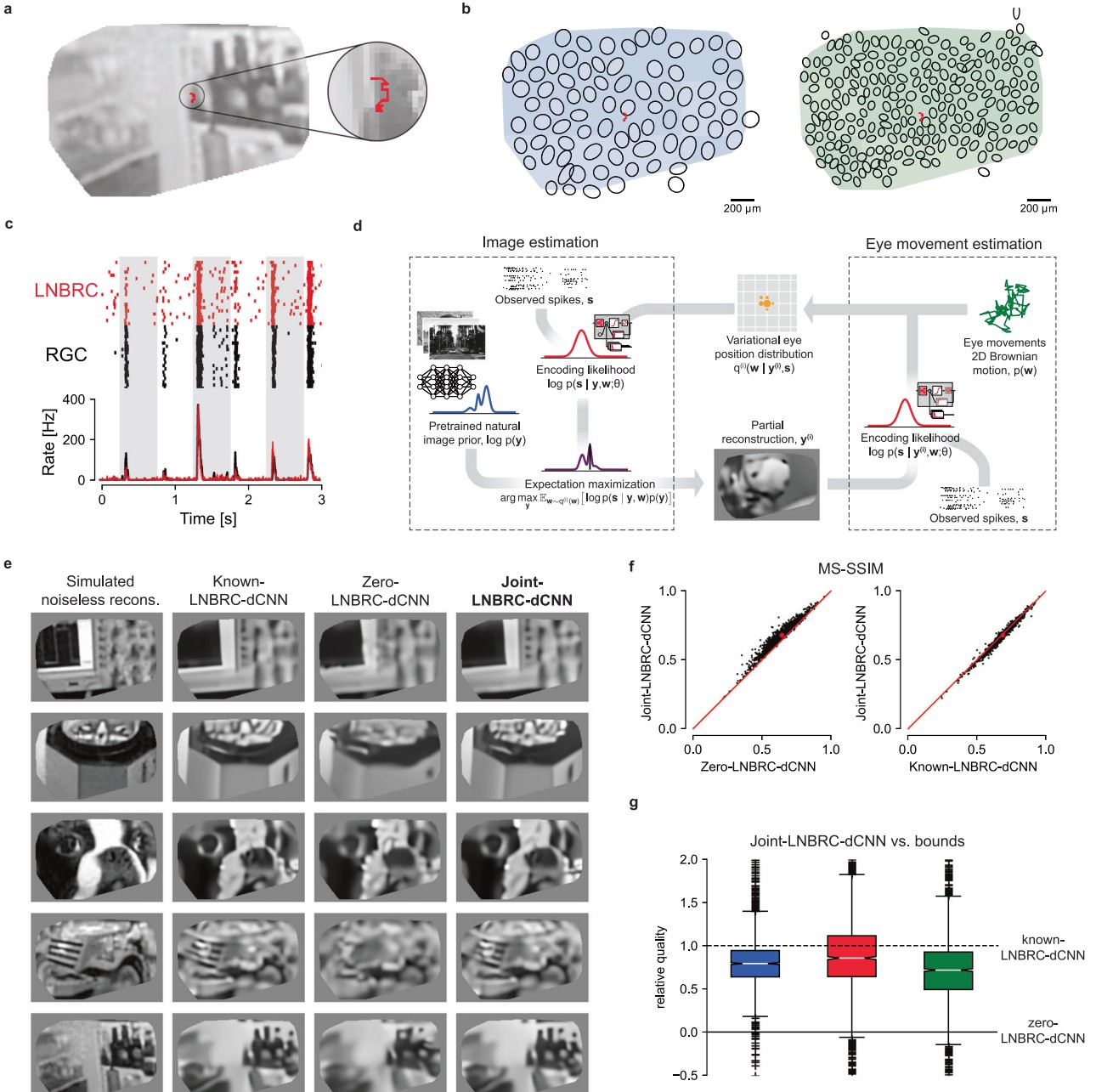

**Fig. 2 | Reconstruction of jittered natural images from RGC spikes. a** Example reconstructed image from simulated noiseless RGC responses, with an example fixational drift eye movement trajectory overlaid (red). The original stimulus is from the ImageNet dataset[11] and is not displayed due to copyright restrictions. **b** Example ON parasol cell receptive field mosaic (left) and ON midget cell mosaic (right), with drift trajectory (red). The simulated eye movements were typically comparable to the size of a midget RGC receptive field. **c** Top: comparison of spikes recorded from an ON parasol RGC over repeated presentations of the same stimulus (black ticks) to simulated responses of the fitted LNBRC model (red ticks). Bottom: average spike rates over time corresponding to the above rasters. **d** Schematic of joint-LNBRC-dCNN reconstruction. The algorithm alternates between an image estimation update step (left), in which the stimulus is reconstructed using the LNBRC model and denoiser CNN image prior to maximizing the expected log-posterior over a variational distribution for eye movements, and an eye movement update step (right), in which the variational distribution for eye movements is updated given the reconstructed image. **e** Example reconstructions, using LNBRC encoding model and dCNN prior. Columns: Simulated noiseless reconstruction, a reconstruction of the stimulus from linear projections onto the

LNBRC filters (see the "Methods" section); Known-LNBRC-dCNN, MAP reconstruction with known eye movements; Zero-LNBRC-dCNN, MAP reconstruction with the (incorrect) assumption of zero eye movements; and Joint-LNBRC-dCNN, joint estimation of image and eye movements. The original stimuli were taken from the ImageNet dataset[11] and are not displayed due to copyright restrictions. The complete comparisons are available at https://github.com/wueric/wu-nature-comms-2024. **f** Left: Performance of joint-LNBRC-dCNN vs. zero-LNBRC-dCNN. Reconstruction quality using joint-LNBRC-dCNN exceeded that of zero-LNBRC-dCNN for nearly every image. Right: performance of joint-LNBRC-dCNN vs. known-LNBRC-dCNN. Known-LNBRC-dCNN algorithm slightly outperformed joint-LNBRC-dCNN. **g** Relative reconstruction quality for the joint estimation procedure joint-LNBRC-dCNN, normalized for each image to zero-LNBRC-dCNN (solid line) and known-LNBRC-dCNN (dashed line). The boxes mark the median and the inter-quartile range (IQR), while the whiskers extend to 1.5 times the IQR. Outliers are marked with a +. For all three preparations, the relative reconstruction quality for joint-LNBRC-dCNN was typically near 1 (mean: 0.976, 1.02, and 0.793), the performance with known eye movements.

motion[23,27,34,35] with diffusion constant in the range expected in humans[23,37] and macaques [Z.M. Hafed and R.J. Krauzlis, personal communication, June 2008] (see the "Discussion" section).

The LNBRC model was fitted to RGC responses to these stimuli by maximizing likelihood. Model fit quality was assessed by comparing the model-simulated spikes with recorded data (Fig. 2c), and by computing the fraction of response variance explained by the model. Although some small systematic deviations from the data were observed (Fig. 2c), in general, the LNBRC model effectively captured responses to natural stimuli with fixational eye movements (Fig. S1).

The fitted LNBRC was combined with the dCNN natural image prior to perform the simultaneous estimation of the stimulus image and eye position using a modified approximate MAP procedure. To avoid computationally expensive marginalization over the eye movement trajectories, an expectation-maximization (EM) algorithm[27] was used to alternate between reconstructing the intermediate image that maximized the expected log posterior over an estimated distribution of eye movement trajectories and using that intermediate image to update the eye movement distribution (Fig. 2d, also see the "Methods" section and Supplementary Information).

The effectiveness of this procedure (labeled joint-LNBRC-dCNN) in compensating for unknown eye movements was evaluated by comparing reconstruction quality to the case in which eye movements were known exactly (known-LNBRC-dCNN), and the case in which eye movements were incorrectly assumed to be zero (zero-LNBRC-dCNN). Reconstruction quality for joint-LNBRC-dCNN exceeded that of zero-LNBRC-dCNN (mean MS-SSIM of 0.677, 0.652, and 0.638 for each preparation for joint-LNBRC-dCNN, in comparison with 0.642, 0.617, and 0.615 for zero-LNBRC-dCNN for the same preparations) and approached that of known-LNBRC-dCNN (mean MS-SSIM of 0.685, 0.656, and 0.646 for the same preparations). Notably, this was the case for nearly every image evaluated, for every preparation (Fig. 2f, g). Qualitative comparisons (Fig. 2e) revealed that the joint solution recovered substantially more image structure and fine spatial detail than the one that ignored eye movements, and produced reconstructions that were similar in content and quality to those produced with known eye movements. These results demonstrate that compensation for fixational drift eye movements is critical for recovering fine spatial detail in the visual scene and that the RGC spikes alone are sufficient to perform this compensation without direct access to eye position information.

**Fixational eye movements enhance the retinal visual signal**
To test whether fixational drift eye movements improve or degrade retinal coding of natural images, reconstruction quality was examined as a function of realized drift eye movement magnitude, quantified as the standard deviation of the random displacement occurring during each trial. In all three preparations, when simultaneously estimating both the image and eye positions, the mean reconstructed image quality increased with increasing magnitude of eye movements over nearly the entire naturalistic range tested (Fig. 3a, solid). The same was true when reconstructing with known eye positions (Fig. 3a, dashed), demonstrating that the improvement was due to an improved retinal signal. Validation with the LPIPS perceptual distance measure[38] yielded similar results (Fig. S4). Thus, fixational drift eye movements enhance, rather than degrade, the retinal representation.

Interestingly, the enhancement occurred even though the fixational drift was small: the drift magnitudes tested were comparable to the size and spacing of midget cell receptive fields in these recordings (Fig. 3; see the "Discussion" section). Specifically, the standard deviation of the random displacement in the largest drift trajectories tested was ~30 μm (Fig. 3), and the peak excursion of these large drift trajectories ranged from 31.1 to 73.8 μm (5th and 95th percentiles). By comparison, the median midget RGC receptive field radii were 36.4, 40.7, and 30.7 μm for the blue, red, and green preparations, respectively (Fig. 3), and midget cell spacing is approximately twice the

receptive field radius[39]. Thus, although the larger drift trajectories moved image features from one midget cell receptive field to another, more frequently, the image displacements were smaller, with unknown implications for the corresponding foveal image reconstruction problem (see the section "Discussion"). Note that the typical spacing of cone photoreceptors at the eccentricities of these recordings (~14 μm)[40] was small compared to the image displacements.

The benefits of fixational drift could, in principle, arise from an overall increase in spike rates, because RGCs are responsive to intensity changes over time, which increase in the presence of drift. Indeed, the mean number of spikes increased with increasing eye movement magnitude: in the three preparations, the mean spike rate increased by 3.1%, 2.4%, and 2.1% for each 10 μm increase in eye movements random displacement standard deviation, with Pearson correlation coefficients of 0.97, 0.98, and 0.76, respectively. Thus, at least some of the improvement in reconstructed image quality may be attributable to increased RGC firing.

Another possibility is that image reconstruction is enhanced by a more accurate estimation of the eye movement trajectory with larger eye movements. This did not appear to be the case: the accuracy of eye trajectory reconstruction declined with increasing magnitude of drift, albeit much more slowly than for the model that assumed zero movement (Fig. 3b). Thus, the improved image reconstruction with increasing magnitude of eye movements was attributable to a more faithful encoding of the stimulus in RGC spikes rather than a more precise implicit signal about eye position.

The potentially distinct impacts of fixational drift on each of the parasol and midget RGC signals were examined by reconstructing one population at a time. Though midget RGCs have smaller receptive fields and are associated with fine pattern vision, their slower stimulus temporal integration could cause greater positional uncertainty in the presence of drift eye movements, potentially degrading the quality of the representation compared to that of the parasol RGCs. However, this was not the case. Midget-only reconstructions had systematically higher quality than parasol-only reconstructions and contained greater fine spatial detail (Fig. 3c, also Fig. S5), demonstrating that midget cells encoded a greater fraction of the stimulus than parasol cells. Reconstruction quality improved with increasing realized drift magnitude for both the parasol-only and midget-only reconstructions. While the error in estimated eye position increased much more slowly for each population than if eye movements were ignored (Fig. 3d), showing that both cell groups were informative of the eye movement trajectory, the position error was substantially smaller in the midget-only reconstructions, suggesting that midget RGCs were largely responsible for encoding fine eye movements.

**Fixational eye movements evoke more precisely timed spikes**
Previous work in the turtle retina has revealed greater temporal precision of RGC spikes in the presence of simulated fixational eye movements than in a still image[41]. To test whether this precision could enhance natural image reconstruction, the observed RGC spikes were randomly perturbed in time by intervals drawn from Gaussian distributions with increasing standard deviation (1, 2, 5, 10, 20, and 40 ms), and reconstructions were computed with the perturbed spikes. To ensure optimal reconstruction with the perturbed spikes, the LNBRC models used for estimating likelihood were refitted to perturbed data. Spike time perturbation had two effects on the retinal signal. First, it disrupted the spike train temporal structure, resulting in reduced strength of the fitted LNBRC feedback filter (Fig. S9). Second, because the spike times of each cell were shifted independently, it spread out the spiking synchrony between neighboring cells in time, resulting in reduced peak amplitudes of the fitted LNBRC coupling filters (Figure S8). For the flashed stimuli, reconstruction quality declined gradually for spike time perturbations up to about 10 ms, and then declined more sharply for larger perturbations, indicating that spike time

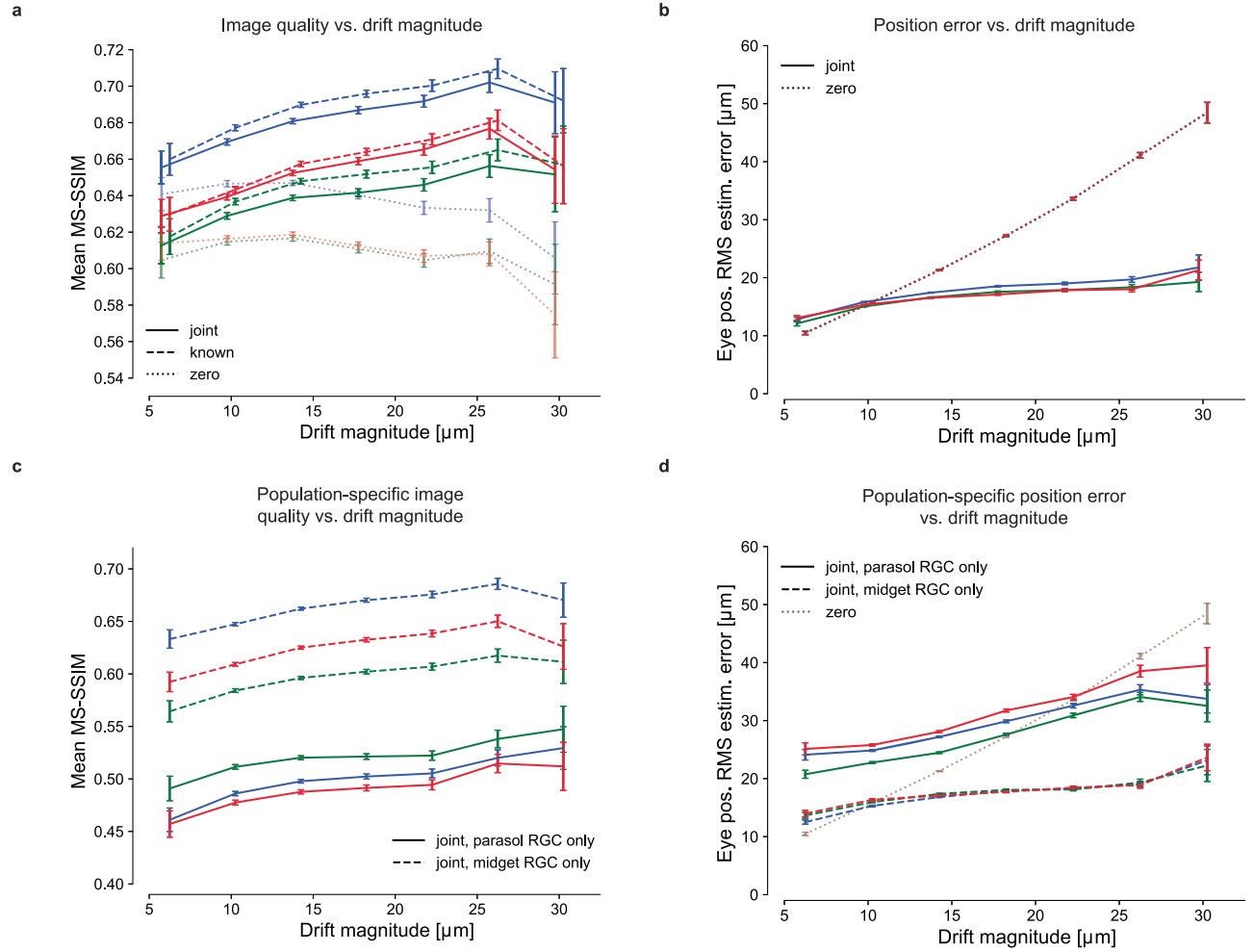

**Fig. 3 | Effects of fixational drift magnitude on reconstruction quality. a** Image reconstruction performance for three preparations (delineated with different colors) as a function of the magnitude of eye movements simulated during the stimulus presentation, for joint-LNBRC-dCNN (solid line), known-LNBRC-dCNN (dashed line), and zero-LNBRC-dCNN (dotted line). The magnitude of drift eye movements was quantified as the standard deviation in the eye position occurring during each trial. The error bars in every panel correspond to the standard error of mean reconstruction quality. In all preparations, reconstruction quality for joint-LNBRC-dCNN and known-LNBRC-dCNN increased with eye position jitter, up to (but not including) the largest eye movements evaluated. Reconstructions for zero-LNBRC-dCNN were less accurate than both known-LNBRC-dCNN and joint-LNBRC-dCNN and further decreased with increasing eye movements. **b** Eye position estimation error as a function of the magnitude of movement, for the same experimental preparations.

When eye movements were ignored (zero-LNBRC-dCNN, dotted line), the error in estimated eye position increased linearly, as expected with a 2D Brownian motion. When eye movements were jointly estimated (joint-LNBRC-dCNN; solid lines), the error increased, but more gradually. **c** Parasol-only (solid line) and midget-only (dashed line) joint-LNBRC-dCNN image reconstruction performance as a function of the magnitude of movement, for the same experimental preparations. In all preparations, reconstruction quality increased with increasing magnitudes of fixational drift for both parasol-only and midget-only reconstructions. Midget-only reconstructions had systematically better quality than parasol-only reconstructions in all preparations. **d** Parasol-only (solid line) and midget-only (dashed line) eye position estimation error, for the same experimental preparations. For both parasol-only and midget-only reconstructions, the eye position estimation error increased more slowly than if eye movements were ignored (dotted line).

structure finer than 10 ms was relatively unimportant (Fig. 4a). However, for stimuli with simulated fixational drift, reconstruction quality deteriorated more rapidly as a function of spike time perturbation, and was affected more than the flashed reconstructions by perturbations on the order of 5 ms (see the "Discussion" section). This was true regardless of whether eye movements were jointly estimated (Fig. 4b, solid lines) or known a priori (Fig. 4b, dashed lines). Repeating the analysis with the LPIPS perceptual distance measure yielded similar results (Fig. S6). Thus, eye movements encode the spatial structure of natural images into the fine temporal structure of spikes, and exploiting this aspect of retinal encoding enhances image reconstructions.

### Correlated firing between RGCs contributes to reconstructed image quality

Although previous work[5,42] has demonstrated that correlated firing of RGCs affects the transmitted information for simple stimuli, the

importance of such correlations for naturalistic stimuli is less certain[43–45] as are the distinct roles of stimulus-dependent (signal) and stimulus-independent (noise) correlations. To better understand the role of correlations in naturalistic visual signaling by the retina, two analyses were performed. First, image reconstruction was performed with a readout that ignored all correlations. Second, images were reconstructed from synthetic data created by shuffling the recorded responses of each cell across repeated presentations of the same stimulus. These analyses are presented below in turn.

To probe the role of correlations in reconstructions, LNBR ("uncoupled") encoding models were fitted to the experimental data, and the resulting natural image reconstructions were compared to the results obtained with the full LNBRC ("coupled") model, similar to a previous analysis performed with white noise stimuli[5]. The uncoupled models lacked the ability to represent correlated firing between RGCs beyond linear filtering of the shared visual stimulus and were fitted and

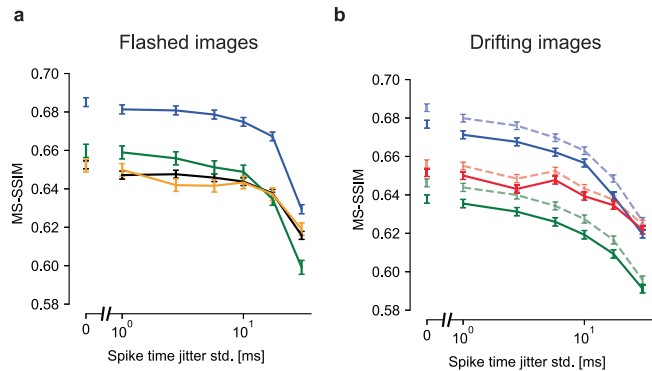

**Fig. 4 | Effects of spike timing precision on reconstruction quality.**
**a** Reconstruction performance for flashed images as a function of spike timing perturbation in four experimental preparations (colors). The x-axis is plotted on a log scale, with a broken axis to facilitate comparison with temporally unperturbed data. Error bars in all panels correspond to the standard error. Performance at each level of temporal perturbation was evaluated using $N = 1500$, $N = 1750$, $N = 750$, and $N = 750$ images for the blue, black, green, and yellow preparations, respectively. Reconstruction degraded modestly up to spike time perturbations of ~10 ms.
**b** Reconstruction performance for the fixational drift stimulus. Blue and green lines correspond to their color-matched preparations in (**a**). The dashed and faded lines correspond to estimation with known eye movement trajectories and the solid lines to joint estimation of the image and eye trajectory. Performance at each level of temporal perturbation was evaluated using $N = 1992$ images for each experimental preparation. Regardless of whether the eye movement trajectories were known or jointly estimated, performance declined smoothly starting at a temporal jitter of ~2–5 ms.

used to compute reconstructions in an identical manner to the coupled models. For both the flashed and the fixational drift stimuli, the reconstructions computed using the coupled models were significantly more accurate than those computed using the uncoupled models. For the flashed stimuli (Fig. 5a), the mean MS-SSIM differences between coupled and uncoupled reconstructions were 0.023, 0.024, 0.037, and 0.023, (all p-values $< 1 \times 10^{-10}$, coupled > uncoupled, Wilcoxon signed rank test, $N = 1500$, $N = 1750$, $N = 750$, and $N = 750$, respectively), and for the fixational drift stimuli (Fig. 5b) the differences were 0.019, 0.010, and 0.039 (all p-values $< 1 \times 10^{-10}$, coupled > uncoupled, Wilcoxon signed rank test, $N = 1992$ for each). Thus, for naturalistic stimuli, knowledge of correlated firing properties of RGCs beyond that which could be explained by linear filtering of the shared stimulus was necessary to effectively decode image content.

The impact of correlated firing on natural image reconstruction could not be attributed to noise correlations alone, in contrast to what was seen in prior work using white noise stimuli[5]. While the cross-correlograms simulated with the coupled LNBRC model (Fig. 5e, red) accurately matched both real data (black) and data shuffled across repeats to remove noise correlations while preserving the average properties of each cell (blue), the cross-correlograms simulated with the uncoupled LNBR model (green) often differed markedly from both. This indicates that the coupled model better represented signal correlations in RGC firing than the uncoupled model, an unexpected finding with implications for retinal modeling (see the "Discussion" section). The coupled model also explained a systematically greater fraction of firing variation than the uncoupled model (Fig. 5f).

To probe whether noise correlations contributed significantly to the retinal signal, reconstructions were compared with the recorded responses of each cell shuffled across repeated stimulus presentations. Using the LNBRC fitted to the unshuffled data (i.e. with full knowledge of noise correlations), reconstructions were obtained for both the real (unshuffled) repeats as well as the shuffled data. For the flashed stimuli, the reconstructions computed from unshuffled spikes were marginally more accurate than those computed from the shuffled spikes, for all

preparations tested, with mean difference values of $7.4 \times 10^{-3}$, $6.6 \times 10^{-3}$, $5.1 \times 10^{-3}$, and $3.4 \times 10^{-3}$ (all p-values $< 1 \times 10^{-10}$, data > shuffled, Wilcoxon signed rank test, $N = 150$ for all) respectively. For the fixational drift stimuli, the effect was similar: the difference was significant for two of the three preparations tested, with mean values $5.9 \times 10^{-5}$, $9.5 \times 10^{-4}$, and $7.6 \times 10^{-3}$ (p-values 0.45, 0.017, and $< 1 \times 10^{-10}$, data > shuffled, Wilcoxon signed rank test, $N = 149$ for all). While statistically significant, the effect was substantially smaller than that of removing the coupling filters, suggesting that the contributions of noise correlations to the retinal representation of natural stimuli were modest. Analysis using the LPIPS perceptual distance measure yielded similar results (Fig. S7). Furthermore, a comparison of the raw and shuffled repeat cross-correlograms (black and blue lines in Fig. 5e for data and shuffled, respectively) and cross-correlogram peak height (Fig. 5h) showed that noise correlations were substantially smaller than signal correlations. These results demonstrate that noise correlations contributed substantially less than signal correlations to retinal representations of naturalistic scenes, a striking difference compared to reconstruction performed previously using white noise stimuli[5] (see the "Discussion" section).

## Discussion

We have presented a Bayesian method to invert the retinal code, reconstructing visual images from the measured spiking responses of populations of RGCs, and used these reconstructions to understand the effect of fixational drift on the retinal signal. The reconstruction process is not intended as a model of how the brain processes visual images[46], but as a tool for making explicit the content of the retinal signal in the form of an image, providing insight into the sensory content that is available in neural activity and the way this content is represented[4].

These analyses relied on both the performance and interpretability of the reconstruction method, leveraging both the sophistication of and separation between the likelihood and prior models. The likelihood, obtained from an LNBRC encoding model, effectively captured RGC responses to naturalistic stimuli with modular components that represented stimulus dependency, spike history dependence, and spike time correlations. Although it is not matched to the details of biological circuitry or cellular biophysics[47,48], it is convex in its parameters, and thus is reliably fitted to spiking data and is computationally feasible for use in MAP image reconstruction. Separately, natural image structure was captured using the prior implicit in a neural network trained to denoise images. Such implicit priors, related to the "score-based generative models" or "diffusion models" that have recently emerged in the machine learning community, offer unprecedented power for capturing image properties while requiring relatively modest amounts of training data[15–17]. Most importantly, the likelihood and prior components may be combined to support a Bayesian formulation, which offers enhanced interpretability because the two components can be independently altered to evaluate their contributions to the retinal representation.

Image reconstruction revealed that the retinal signal alone is sufficient for accurately decoding visual stimuli in the presence of unknown fixational eye movements, consistent with previous theories[26,34,35,49] and psychophysical studies[25,28,30,50]. Though previous computational investigations[27,34,35] have explored this possibility in simulation with simplified stimuli, the present work tested the idea empirically with physiologically recorded RGC spikes and naturalistic stimuli. Of course, our findings do not exclude the possibility of additional extra-retinal signals that could help to compensate for fixational eye movements, as has been reported previously[31]. Indeed, the small gap in quality between images reconstructed by the joint algorithm and those reconstructed with full knowledge of the eye position suggests the potential benefits of incorporating extra-retinal signals.

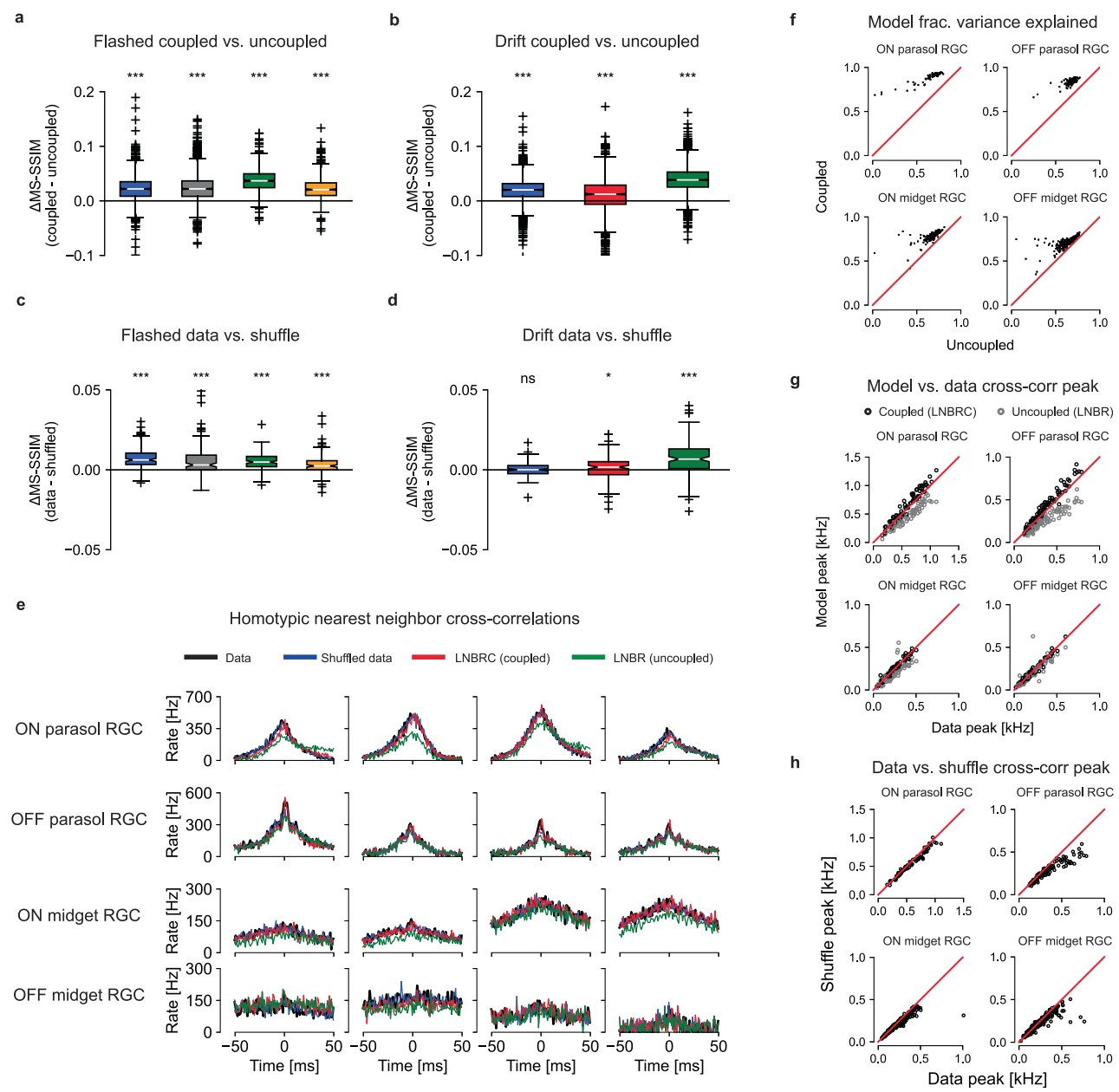

**Fig. 5 | Effects of coupling. a** Differences in reconstruction quality between the coupled model (LNBRC) and uncoupled model (LNBR) for flashed natural images. Mean differences for four preparations: 0.023, 0.024, 0.037, and 0.023 ($p$-values $3.7 \times 10^{-184}$, $1.2 \times 10^{-211}$, $8.6 \times 10^{-122}$, and $8.5 \times 10^{-111}$, respectively, Wilcoxon one-sided ranked sign test). For panels (a-d), the box marks the median and the inter-quartile range (IQR), while the whiskers extend to 1.5 times the IQR. Outliers are marked with a +. **b** Same as (**a**), for reconstruction with fixational drift using the joint approach. Mean differences for three preparations: 0.019, 0.010, and 0.039 ($p$-values $3.2 \times 10^{-218}$, $4.2 \times 10^{-62}$, and $2.5 \times 10^{-307}$, respectively). The blue and green boxes correspond to the same experimental preparations as the blue and green boxes in (**a**). **c** Differences in reconstruction quality between the unshuffled and shuffled trials for flashed image reconstructions, using LNBRCs fitted to unshuffled data. Mean differences: $7.4 \times 10^{-3}$, $6.6 \times 10^{-3}$, $5.1 \times 10^{-3}$, and $3.4 \times 10^{-3}$ ($p$-values $4.5 \times 10^{-23}$, $5.6 \times 10^{-12}$, $3.1 \times 10^{-20}$, and $3.4 \times 10^{-12}$, respectively), substantially smaller than those in (a). (**d**) Same as (c), for reconstruction with fixational drift using the

joint approach. Mean differences (left-to-right): $5.9 \times 10^{-5}$, $9.5 \times 10^{-4}$, and $7.6 \times 10^{-3}$ ($p$-values 0.45, 0.017, and $9.1 \times 10^{-16}$), substantially smaller than those in (**b**). **e** Example homotypic (same cell type) nearest-neighbor spike train cross correlograms. Panels **e**–**h** are computed using the blue experimental preparation in panels (**a**–**d**) using repeat presentations of jittered natural image stimuli. Cross-correlograms for the data are shown in black, and repeat-shuffled data in red. Simulated cross-correlograms for the LNBRC (coupled) models and for the LNBR (uncoupled) models are shown in red and green, respectively. **f** Fraction of PSTH variance explained by the coupled and uncoupled models. **g** Cross-correlogram peak height comparison between model simulations and data. While the LNBRCs sometimes overestimated the correlations in the data, the LNBRs systematically underestimated them. **h** Cross-correlogram peak height comparison between repeat data and shuffled repeat data, for the same preparation as (**f**, **g**), with the jittered stimulus. Except for the OFF parasol cells, peak heights were similar, indicating that noise correlations were only weakly present.

Increased fixational drift systematically improved reconstruction quality. This provides additional evidence in support of the theory that fixational drift serves a useful function in visual processing, modulating high-frequency spatial detail into the time domain[22,23,33,51] and/or enabling super-resolution positional sampling[25,27]. Furthermore,

because this held even when the eye movements were unknown a priori, it demonstrates that fixational drift specifically improves the fidelity of the retinal representation of natural images.

Because the present work analyzes recordings performed in the peripheral retina, the findings may not be directly applicable to foveal

vision. Foveal RGCs are denser and have smaller receptive fields than peripheral RGCs, and the density of cone photoreceptors is more than an order of magnitude higher in the fovea than in the periphery. As a result, fixational drift eye movements sweep the visual scene across 15–20 RGC receptive fields in the fovea[40,52], in comparison to the 1–2 RGC receptive fields in the peripheral retina, and also across many more cones. The computational challenges of reconstructing the visual scene accurately from RGC responses may therefore differ in the fovea and periphery, with foveal reconstructions incorporating input signals from many more RGCs. A direct test of the implications of the present work for central vision will require high-quality recordings of RGC populations in the fovea, which are beyond the reach of current experimental techniques. However, the computational methodology described here offers a path to exploring the effects of drift eye movements in the central retina when such recordings become feasible.

The impact of fixational drift on the fidelity of the retinal representation may be affected by aspects of the visual stimulus design. For simplicity, drift eye movements were simulated using 2D Brownian motion, ignoring the effects of persistence and self-avoidance[53,54] observed in biological eye movement drift trajectories. These simplifications are unlikely to significantly affect the overall findings, as the joint reconstruction algorithm makes few assumptions about the structure of drift eye movements. As the trajectories generated by Brownian motion are more random than physiological drift trajectories reflecting these properties, the inclusion of additional constraints associated with these properties of eye movements would be expected to improve reconstructed image quality. In addition, our stimuli simulated saccades by abruptly switching between different images. This crude approximation[55] potentially evokes a stronger transient RGC response than real saccades would. Because the transient response carries substantial image information, our conditions may underestimate the impact of drift eye movements on reconstructed image quality.

Precisely timed spikes were shown to play an important role in the retinal representation of natural images with fixational drift. Though RGCs can spike with temporal precision on the order of 1 ms[56–59], previous studies have shown that integration times on the order of 10 ms provide the highest-fidelity readout of steady visual motion from RGCs[60,61]. Consistent with these studies, and with previous flashed natural image reconstruction[2,3], the present findings showed that flashed image reconstruction was robust to spike train temporal perturbations up to 10 ms. However, in the presence of fixational drift, finer temporal precision (2–5 ms) was required for optimal reconstruction. This is consistent with work suggesting that the spike train temporal structure induced by fixational eye movements encodes high-frequency spatial detail[22,41,62] and motion[41,63].

As in previous work on the reconstruction of white noise stimuli[5,42], correlated RGC firing contributed significantly to reconstructing natural images with simulated fixational drift (but see ref. 44). Surprisingly, however, in the present work, the effect was primarily attributable to stimulus-driven correlations rather than the noise correlations that dominated the results in the prior work. The weak role of noise correlations in the present data matched the results obtained by reconstructing flashed natural images using more limited approaches[2,3] and results from decoding dynamically varying artificial movies[64]. The observed importance of signal correlations may also explain the observation that the LNBR (uncoupled model) provides a limited explanation of natural visual signals in individual RGCs[65].

Future work could extend the Bayesian reconstruction framework to characterize the function of spatio-temporal nonlinearities in the retinal representation of naturalistic stimuli. Though recent work with subunit[66–68] and neural network[69] encoding models has demonstrated substantial improvements in accounting for RGC spiking, the roles of the spatio-temporal nonlinearities contained in these models remain

unclear. Combining such encoding models with improved image priors to draw reconstruction samples from the posterior[17,70,71] could further reveal the interplay between retinal coding and natural image statistics.

## Methods

### Multi-electrode array recordings

Preparation and recording methods are described in detail elsewhere[3,7,60,72]. In brief, eyes were enucleated from terminally anesthetized macaque monkeys used by other laboratories, in accordance with Institutional Animal Care and Use Committee requirements. All animals were handled according to approved institutional animal care and use committee (IACUC) protocols (#28860) of Stanford University. The protocol was approved by the Administrative Panel on Laboratory Animal Care of Stanford University (Assurance Number: A3213-01). Segments of peripheral (7–17 mm eccentricity, 6–12 mm temporal equivalent eccentricity, or 29–56°[60]) RPE-attached retina ~3 mm in diameter were cut from the eye in dim light, and placed RGC side down on a multi-electrode array (MEA). A 512-channel MEA system with 60 μm pitch between electrodes and a 2 × 1 mm rectangular recording area was used to perform the recordings[7]. This system bandpassed and digitized the raw recorded voltage traces at 20 kHz. The preparations were perfused with Ames' solution (30–34 °C, pH 7.4) bubbled with 95% $O_2$ and 5% $CO_2$ throughout the recordings. In total, retinas from five animals were used (ages 12, 15, 12.5, 15–20, and 18 years, all male).

Spike sorting was performed with YASS[73]. RGCs of the four numerically dominant types in macaque (ON parasol, OFF parasol, ON midget, OFF midget) were identified manually based on receptive fields and autocorrelation functions characterized with a spatio-temporal white noise stimulus according to previously described procedures[10] and were matched to spike-sorted units from the natural scenes recordings by matching electrical images (voltage templates). Only identified RGCs of the four major cell types were used in the analysis. The four preparations used for the flashed reconstructions contained 691, 592, 704, and 677 total cells, and the three preparations used for the jitter eye movements reconstructions contained 715, 604, and 775 total cells.

### Visual stimuli

Visual stimuli were presented on a 120 Hz, gamma-corrected CRT monitor (Sony Trinitron Multiscan E100) that was optically reduced and projected onto the retina. The 120 Hz refresh rate used in these experiments was sufficient to exceed the temporal resolution of the parasol cells. All experiments occurred at low photopic light levels (2000, 1800, and 800 isomerizations per second for the L, M, and S cones, respectively, at 50% illumination; see refs. 72,74). The visual stimulus covered an area of 3.5 × 1.75 mm, extending well beyond the recording area of the MEA.

A 30-min spatio-temporal white noise stimulus was used to identify RGCs of the major cell types and to characterize the locations of their receptive fields[75]. The stimulus consisted of a grid of pixels (either 44 or 88 μm in size), whose intensities were drawn independently and randomly from a binary distribution. The stimulus was refreshed at either 30 or 60 Hz.

Flashed natural images from the ImageNet database[11,12] were presented to the retina according to ref. 3. Images were converted to grayscale, cropped to 256 × 160 resolution, and padded with gray borders. The stimulus extended beyond the boundaries of retinal preparation and fully covered all receptive fields. Each pixel in the image measured ~11 × 11 μm when projected on the retina. Each image was displayed for 100 ms (12 frames at 120 Hz), and sequential images were separated by a 400 ms uniform gray screen.

The natural movies with simulated fixational eye movements consisted of ImageNet images presented for 500 ms each (60 frames at

120 Hz), with no gray screen separation. For each image, eye movements were simulated by shifting the image during each frame transition according to a discretized 2D Brownian motion with a diffusion constant of 10 μm²/frame, consistent with estimates of fixational eye movements in both human[23,37] and non-human primates [Z.M. Hafed and R.J. Krauzlis, personal communication, June 2008]. Simulated eye movements were drawn independently of the image. The movies were presented in sequence, with no gray screen between movies.

The receptive fields of the recorded RGCs covered only a central region of the stimulus field, leaving a perimeter region for which no cells were recorded. To evaluate image quality only over regions of the stimulus corresponding to recorded cells, a valid region was constructed, consisting of the convex hull of the receptive fields of the full RGC population. Only pixels in this valid region were used to compute image quality.

## Fitting LNBRC models of RGC spiking

The linear-nonlinear-Bernoulli with recursive coupling (LNBRC) is a modified form of the GLM model developed in ref. [5]. It generalizes the classical linear-nonlinear-Poisson (LNP) spiking model by incorporating recursive feedback (spike history) and neighboring cell coupling filters to capture spike train temporal structure and cell-to-cell correlations (Fig. 1b). For RGC $i$, the LNBRC has the following parameters: (1) $\mathbf{m}_i$, the linear spatio-temporal stimulus filter; (2) $f_i[t]$, the recursive feedback filter; (3) $c_i^{(j)}[t]$, the coupling filters to neighboring RGCs indexed by $j$, where neighboring cells were included if their receptive field centers fell within twice the median nearest neighbor distance for parasol cells and 2.5 times the median nearest neighbor distance for midget cells; and (4) $b_i$, an additive bias. Let $\mathbf{v}[t]$ denote a temporal window of the visual stimulus movie up to and including time $t$, $*$ a time-domain convolution, $s_i$ the spike train of cell $i$, and $N_i$ the set of cells coupled to cell $i$. The instantaneous spiking probability for cell $i$ is computed from the generator signal, $g_i[t]$:

$$g_i[t] = \mathbf{m}_i^T(\mathbf{v}[t-1]) + (s_i * f_i)[t-1] + \sum_{j \in N_i}(s_j * c_i^{(j)})[t-1] + b_i \qquad (1)$$

Temporal filters in the LNBRCs were strictly causal so that the firing probability at time $t$ depended only on the visual stimulus and observed spikes occurring strictly before time $t$. Time was discretized in 1 ms bins, corresponding approximately to the duration of the refractory period of a neuron. Since at most one spike could occur in each time bin, a Bernoulli random process was used to model spiking, with a sigmoidal nonlinearity of the form $e^x/(1+e^x)$ mapping the generator signal to an instantaneous firing probability, resulting in the encoding negative log-likelihood

$$-\log p(\mathbf{s}|\mathbf{v}) = \sum_t [\log(1 + \exp\{g_i[t]\}) - s_i[t]g_i[t]], \qquad (2)$$

which is jointly convex in the model parameters. The stimulus filter was assumed to be space-time separable (rank 1), and the spatial component was cropped to a rectangular region surrounding the cell's receptive field and represented in terms of a 2D cubic spline basis[76]. The feedback, coupling, and temporal components of the stimulus filter were each parameterized as linear combinations of low-rank 1D raised cosine basis functions[5].

The models were fitted to recorded RGC spikes by maximizing the parameter likelihood and were regularized with an $L_1$ penalty to induce sparsity on the spatial component of the stimulus filter and an $L_{2,1}$ group-sparsity penalty on the cosine basis weights of the coupling filters to eliminate unnecessary cell-to-cell coupling. Because of the assumed space–time separability of the stimulus filter, the LNBRCs were fitted using coordinate descent, alternating between solving a convex minimization problem for the stimulus spatial filter, feedback filter, coupling filters, and bias, and solving a convex minimization problem for the stimulus time course filter, feedback filter, coupling filters, and bias. All optimization problems were solved using FISTA[77], an accelerated proximal gradient method, using the formulation for the $L_{2,1}$-regularized problem presented in ref. [78]. Optimal values for the weights controlling the strength of the $L_1$ and $L_{2,1}$ regularizers were found using a grid search to minimize the average test negative log-likelihood over four randomly chosen cells of each cell type. Within each preparation, every RGC of a given type used the same hyperparameters.

The LNBRCs were fitted separately for each cell and required about 180 s of compute time per cell for the static stimulus models and 500 s of compute time per cell for the eye movements models on a single NVIDIA V100 GPU with 32 GB of VRAM.

## LNBRC simulated spike train generation

Simulated spike trains for evaluating model fit quality (Fig. S1) were generated from the LNBRC by computing the value of the generator signal from the stimulus and using simulated Bernoulli random variables to model random spike generation. The recursive feedback contribution to the generator signal was initialized using real observed spike trains, and subsequent generated spike trains were fed back into the model to compute the feedback contribution for future spikes. Because the firing probability computed with the coupled LNBRC was conditional not only on the visual stimulus and simulated cell spiking history but also on the spike trains of nearby coupled RGCs, real spike trains from the experimental data were used to compute the coupling contribution to the generator signal.

## PSTH computation

The peri-stimulus time histogram (PSTH) was computed using RGC responses to repeated presentations of the same visual stimulus by counting the observed spikes within 1 ms time bins, smoothing with a Gaussian kernel with a standard deviation of 2 ms, and then computing the mean over all repeated presentations of the stimulus.

## Fitting benchmark LNP encoding models

Benchmark linear-nonlinear-Poisson (LNP) encoding models were fitted in a similar manner to the LNBRC models. The same spatial basis sets used for the LNBRCs were used for the LNP models. Spikes were counted in 8.33 ms time bins (one bin per stimulus frame). LNP models were parameterized by a spatio-temporal stimulus filter $\mathbf{m}_i$, and a bias $b_i$, resulting in a generator signal of the form $g_i[t] = \mathbf{m}_i^T(\mathbf{v}[t]) + b_i$. An exponential nonlinearity was assumed, resulting in an encoding negative log-likelihood with form

$$-\log p(\mathbf{s}|\mathbf{v}) = \sum_t [\exp g_i[t] - g_i[t]s_i[t]] \qquad (3)$$

which is convex in the LNP model parameters. LNP spatio-temporal filters were assumed to be space-time separable. An $L_1$ penalty was used to induce sparsity in the spatial component of the stimulus filter, and the corresponding weight for that penalty was chosen by performing a grid search with encoding likelihood on the test partition as the objective. Models for each cell were fitted using FISTA[77].

## Reconstruction of flashed images with denoising CNN prior

An iterative Plug-and-Play algorithm[6,20,79] was used to perform MAP reconstruction of flashed static natural images. Rather than solve the MAP problem directly, the algorithm used proximal variable splitting to divide the MAP objective $\arg\min_{\mathbf{y}}\{-\log p(\mathbf{s}|\mathbf{y}) - \lambda \log p(\mathbf{y})\}$ into an encoding sub-problem $\mathbf{x}^{(k+1)} = \arg\min_{\mathbf{x}}\left\{-\log p(\mathbf{s}|\mathbf{x}) + \frac{\rho^{(k)}}{2}|\mathbf{x} - \mathbf{z}^{(k)}|_2^2\right\}$ and a prior sub-problem $\mathbf{z}^{(k+1)} = \arg\min_{\mathbf{z}}\left\{-\lambda \log p(\mathbf{z}) + \frac{\rho^{(k)}}{2}|\mathbf{z} - \mathbf{x}^{(k+1)}|_2^2\right\}$ and iteratively alternated between the two. The encoding sub-

problem was solved using unconstrained convex minimization. The prior sub-problem has the form of a MAP estimation problem for images contaminated with additive Gaussian noise. As such, its solution was approximated using a single forward pass of a convolutional neural network (CNN) pretrained for mean-square-error denoising with specified noise variance $\lambda/\rho^{(k)}$. Ten iterations of alternating optimization were used. $\rho^{(k)}$ was increased per iteration on a log-spaced schedule[6], and hyperparameters $\lambda$, $\rho^{(1)}$, and $\rho^{(10)}$ were found by performing a grid search on an 80-image subset of the test partition with reconstruction MS-SSIM as the objective. A detailed description of the algorithm can be found in ref. 13.

The denoising CNN implicitly representing the natural image prior was implemented using the DRUNet architecture from ref. 6. This CNN had three input channels, consisting of a noisy image, the specified noise variance level, and a binary mask corresponding to the region of the image covered by the recorded RGCs. The CNN was trained using mean square error loss to remove i.i.d. additive Gaussian noise at a variety of noise variance levels from images belonging to the ImageNet database[12].

### Exact MAP reconstruction with 1/F Gaussian prior

Using the 1/F Gaussian prior, the MAP objective had the form

$$\arg\min_{\mathbf{y}} \left\{ -\log p(\mathbf{s}|\mathbf{y}) + \lambda \sum_k |a_k(\mathbf{y})|^2/f_k^2 \right\} \quad (4)$$

where $a_k(\mathbf{y})$ is the amplitude of the Fourier coefficient at frequency $f_k$. Because both the 1/F prior term and the encoding negative log-likelihood are smooth and convex in the image, the MAP problem is an unconstrained convex minimization problem and, hence, was solved with gradient descent. The optimal value of the prior weight $\lambda$ was found with a grid search with reconstruction MS-SSIM as the objective.

### Approximate MAP reconstruction with known eye movements with denoising CNN prior

In the case that the eye movements $\mathbf{w}$ are known a priori, the MAP objective can be simplified into the form

$$\hat{\mathbf{y}} = \arg\max_{\mathbf{y}} \left\{ \log p(\mathbf{s}|\mathbf{y},\mathbf{w}) + \lambda \log p(\mathbf{y}) + \log p(\mathbf{w}) \right\} \quad (5)$$

$$= \arg\max_{\mathbf{y}} \left\{ \log p(\mathbf{s}|\mathbf{y}) + \lambda \log p(\mathbf{y}) \right\} \quad (6)$$

which can be solved using the Plug-and-Play algorithm described above for the flashed case. Hyperparameters were found with a grid search with MS-SSIM as the objective.

### Joint estimation of image and unknown eye movements with denoising CNN prior

The expectation-maximization (EM) algorithm was used to perform MAP estimation for joint estimation of images and eye movements. Letting $\mathbf{w}$ denote the eye movement trajectory over all timesteps, the exact MAP problem with unknown eye movements has form

$$\arg\max_{\mathbf{y}} \left\{ \lambda \log p(\mathbf{y}) + \log \sum_{\mathbf{w}} p(\mathbf{s}|\mathbf{y},\mathbf{w}) p(\mathbf{w}) \right\} \quad (7)$$

which cannot be directly solved because the marginalization over all possible eye movement trajectories $\mathbf{w}$ is intractable. MAP-EM offers an iterative approach for estimating the image $\mathbf{y}$, and consists of alternating steps of (1) finding the image that maximizes the sum of the evidence lower bound and natural image log prior

$$\mathbf{y}^{(i)} = \arg\max_{\mathbf{y}} \left\{ \lambda \log p(\mathbf{y}) + \mathbb{E}_{\mathbf{w} \sim q(\mathbf{w}|\mathbf{y}^{(i-1)},\mathbf{s})}[\log p(\mathbf{s}|\mathbf{y},\mathbf{w})] \right\} \quad (8)$$

over some variational distribution of the eye positions $q(\mathbf{w}|\mathbf{y}^{(i-1)},\mathbf{s})$; and (2) using the resulting estimate of the image $\mathbf{y}^{(i)}$ to update the variational distribution. For computational tractability, we assumed $q$ had form $q \propto p(\mathbf{s}|\mathbf{w},\mathbf{y})r_0(\mathbf{w}_0)\prod_{i=1}^{T}r_i(\mathbf{w}_i|\mathbf{w}_{i-1})$, where $r$ could be an arbitrarily chosen distribution. $q$ was represented approximately using a weighted particle filter with $N=10$ particles. The particle filter was updated once for each frame transition (every 8.33 ms) using a sequential importance resampling procedure[80]. Specifically, at frame $t$, the trajectory represented by each particle was updated by sampling a new eye position from the 2D Gaussian transition probability distribution $p(\mathbf{w}_t|\mathbf{w}_{t-1})$, and then reweighting each particle using the multiplicative weight $\frac{p(\mathbf{s}|\mathbf{w}^{(t)},\mathbf{y}^{(t)})}{p(\mathbf{s}|\mathbf{w}^{(t-1)},\mathbf{y}^{(t-1)})}$ computed using the encoding likelihood model. Mathematical details for the resampling particle filter, including justification for the weight update rule, are provided in the Supplementary Information.

An initial guess for the image $\mathbf{y}^{(0)}$ was reconstructed by assuming a fixed eye position at the origin and performing ten alternating iterations of the algorithm used for the flashed reconstructions. At each intermediate timestep $i$, updated estimates of the image $\mathbf{y}^{(i)}$ were computed by performing a single encoding optimization step $\mathbf{x}^{(i)} = \arg\min_{\mathbf{x}} \left\{ -\mathbb{E}_{\mathbf{w} \sim q(\mathbf{w}|\mathbf{y}^{(i-1)},\mathbf{s})}[\log p(\mathbf{s}|\mathbf{x},\mathbf{w})] + \frac{\rho^{(i)}}{2}|\mathbf{x} - \mathbf{y}^{(i-1)}|_2^2 \right\}$ using unconstrained convex minimization, followed by a single prior optimization step $\mathbf{y}^{(i)} = \arg\min_{\mathbf{z}} \left\{ -\lambda \log p(\mathbf{z}) + \frac{\rho^{(i)}}{2}|\mathbf{z} - \mathbf{x}^{(i)}|_2^2 \right\}$ using a single forward pass of the Gaussian denoiser. To speed computation, images were updated once for every five display frame transitions. Testing on a subset of data indicated that this did not negatively affect reconstruction quality.

### Reconstructions from simulated noiseless linear RGCs

Noiseless linear RGC outputs $\mathbf{s}$ were simulated as linear projections of the image $\mathbf{y}$. This was expressed mathematically as $\mathbf{s} = M^{\mathsf{T}}\mathbf{y}$, where the linear projection filters $M$ are the normalized spatial components of the stimulus filters from the experimentally fitted LNBRC models. $M$ has a rank equal to the number of RGCs. The simulated noiseless linear RGC responses represent an upper bound on the fidelity of the visual signal achieved by the retina, as they ignore both noise in retinal processing and ON/OFF polarity in RGC responses.

Images were reconstructed from simulated noiseless RGC responses using an iterative approximate MAP algorithm, similar to that used for the flashed image reconstructions from recorded data. Because the simulated RGC outputs are noiseless, the likelihood term in the MAP reconstruction objective was converted to a linear equality constraint $\mathbf{s} = M^{\mathsf{T}}\mathbf{y}$, resulting in the constrained non-convex minimization problem

$$\hat{\mathbf{y}} = \arg\min_{\mathbf{y}:M^{\mathsf{T}}\mathbf{y}=\mathbf{s}} \{-\lambda \log p(\mathbf{y})\}. \quad (9)$$

The proximal variable splitting algorithm[6] used to reconstruct flashed images from recorded spike trains was adapted to perform the noiseless reconstructions. Specifically, this algorithm iteratively alternated between solving a constrained encoding sub-problem $\mathbf{x}^{(k+1)} = \arg\min_{\mathbf{x}:M^{\mathsf{T}}\mathbf{x}=\mathbf{s}} \{\frac{\rho^{(k)}}{2}|\mathbf{x} - \mathbf{z}^{(k)}|_2^2\}$ and an unconstrained prior sub-problem $\mathbf{z}^{(k+1)} = \arg\min_{\mathbf{z}} \left\{ -\lambda \log p(\mathbf{z}) + \frac{\rho^{(k)}}{2}|\mathbf{z} - \mathbf{x}^{(k+1)}|_2^2 \right\}$. The encoding sub-problem is convex quadratic minimization with a linear equality constraint, and was solved in closed form as $\mathbf{x}^{(k+1)} = \mathbf{z}^{(k)} - M(M^{\mathsf{T}}M)^{-1}(M^{\mathsf{T}}\mathbf{z}^{(k)} - \mathbf{s})$. The prior sub-problem was solved using a single forward pass of the Gaussian denoiser. Similar to the reconstructions of flashed images from recorded data, ten iterations of alternating optimization were used. $\rho^{(k)}$ was increased per

iteration on a log-spaced schedule[6], and hyperparameters $\lambda$, $\rho^{(1)}$, and $\rho^{(10)}$ were found by performing a grid search on an 80-image subset of the test partition with reconstruction MS-SSIM as the objective.

### Reconstruction quality evaluation
Reconstruction quality was evaluated using Multi-scale Structural Similarity (MS-SSIM)[21], a widely used metric for perceptual similarity. MS-SSIM was calculated over the valid region of the image (described above), ignoring non-informative regions of the stimulus for which no RGCs were recorded. For the jittered reconstructions, the absolute position of the reconstructed image was arbitrary (having been jointly estimated from many jittered input samples), and MS-SSIM was computed for a range of pixel-wise shifts of the reconstructed image, and the best value overall shifts were used.

The results in the paper were also confirmed using the learned perceptual image patch similarity (LPIPS)[38], a more recent measure of perceptual distance computed using pre-trained neural network classifiers. LPIPS has different working principles than MS-SSIM and has been shown to align with human perceptual judgements. Only pixels within the valid region (described above) were used to compute LPIPS.

### Cross-validation data rotation for eye movement analysis
Five-fold data rotation was used to maximize the number of stimulus images available for determining the effect of jitter eye movements on reconstructed image quality. Five different sets of LNBRCs were fitted, each corresponding to distinct and non-overlapping tests and held-out partitions, such that test-quality reconstructions could be produced for nearly every stimulus image presentation in the recorded dataset.

### Cell-type-specific reconstruction analysis
The cell-type-specific analysis was performed by reconstructing the jittered eye movements stimulus using joint-LNBRC-dCNN. For simplicity, the LNBRC models used for this analysis only modeled homotypic correlations, differing from the models used elsewhere in the work. Five-fold data rotation was used for this analysis.

### Spike time perturbation analysis
The spike time perturbation analysis tested the temporal precision of the retinal code by shifting recorded spike times by random amounts drawn from a zero-mean Gaussian, with standard deviations of 1, 2, 5, 10, 20, and 40 ms. To ensure optimal reconstruction at each level of perturbation, the LNBRCs were refitted to each condition. Images were reconstructed using the time-perturbed data and the time-perturbed LNBRCs using the algorithms described above. Optimal hyperparameters were found separately for each time perturbation condition by performing grid searches.

### Uncoupled (LNBR) model correlations analysis
The LNBR (uncoupled) model removes the neighboring cell coupling filters of the LNBRC model, thus losing the ability to represent correlated firing between nearby RGCs, apart from that induced by stimulus correlations. The LNBR is parameterized by a linear spatio-temporal stimulus filter, a recursive feedback filter, and a bias. Using the same notation as in the fully coupled case, the generator signal for cell $i$ in the uncoupled model is written as

$$g_i[t] = \mathbf{m}_i^T(\mathbf{v}[t-1]) + (s_i * f_i)[t-1] + b_i. \tag{10}$$

The LNBRs were fitted with the same 1 ms time bins, sigmoidal nonlinearity, and Bernoulli random spiking model as the LNBRCs. Space-time separability of the stimulus filters was assumed, and the same alternating optimization procedure for fitting was used as in the LNBRC case. An $L_1$ penalty was used to regularize the spatial component of the stimulus filters, and the optimal value of the corresponding hyperparameter was found using a grid search.

Image reconstruction with LNBRs was done in an identical manner as with the LNBRCs. Reconstruction hyperparameters were found using a grid search.

### Noise correlations shuffled repeats analysis
Noise correlations between RGCs were characterized using responses to repeated presentations of the same stimulus. Shuffled responses were constructed by randomly reordering recorded spike trains for each cell across the repeated trials, eliminating noise correlations while preserving single-cell spiking statistics and stimulus-induced correlations. Images were reconstructed for both the real (unshuffled) trials as well as the shuffled trials using LNBRCs fitted to the unshuffled data, using the reconstruction algorithms described above. The change in reconstructed image quality due to shuffling was then computed by taking the mean reconstruction quality across repeats of the same stimulus, and then subtracting the values computed for the shuffled repeats from the values computed for the data repeats.

### Cross-correlogram computation
Cross-correlograms between cells were computed using repeat stimulus presentations by constructing histograms for the differences in spike times of the cells (with 1 ms bins), and taking the mean over all presentations of the same stimulus. Because the stimulus onset and offset frame transitions in the flashed stimuli and transitions between distinct images for the jittered eye movements stimuli induced simultaneous firing of all cells independent of connectivity and shared input structure, a shift predictor correction to the cross-correlograms was applied[81]. This was done by shifting the spike times for the second cell such that the spike trains for that cell corresponded to the response to a different stimulus image, constructing the histogram for the differences in spike times for the cells, and then subtracting said histogram from the original raw cross-correlogram. This removed the component of the cross-correlogram that could be predicted by the trial structure alone, independent of either the spatial content of the stimulus or of noise correlations.

### Reporting summary
Further information on research design is available in the Nature Portfolio Reporting Summary linked to this article.

## Data availability
Source data are provided with this paper. A limited dataset containing spike-sorted spike trains and visual stimuli sufficient for generating the example reconstruction images in the figures has been deposited in a Figshare repository (https://doi.org/10.6084/m9.figshare.23929941). The raw voltage traces are not available due to their large size (>5 TB), the complexity of the data processing pipeline, and ongoing use of the data for unpublished research. Source data are provided with this paper.

## Code availability
The complete source code for demonstrating reconstruction is available on GitHub at https://github.com/wueric/wu-nature-comms-2024. The source code and fitted models are also available in a Figshare repository (https://doi.org/10.6084/m9.figshare.23929941).

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

## Acknowledgements

This work was supported by a National Defense Science and Engineering Graduate (NDSEG) Fellowship (E.G.W.); NSF IGERT 0801700 (N.B.); Wu Tsai Neurosciences Institute Big Ideas (E.J.C.), NSF CRCNS grant 1430348 (E.J.C. and E.P.S.); NEI grants R01EY017992 and R01EY029247 (E.J.C.); and the Simons Foundation (E.P.S.). We thank Zahra Kadkhodaie for the helpful discussions. We thank Fred Rieke and Michele Rucci for their feedback on the work. We thank Corinna Darian Smith and Tirin Moore (Stanford), Jose Carmena and Jack Gallant (UC Berkeley), Jonathan Horton (UCSF), and the California National Primate Research Center for access to primate retinas.

## Author contributions

E.G.W., E.P.S., and E.J.C. conceived the analysis. E.G.W. performed the analysis. N.B. and E.J.C. designed the experiments and visual stimuli. N.B. and C.R. performed the electrophysiological experiments, with help from A.K., A.R.G., and N.P.S. A.S. and A.M.L. provided and supported the multi-electrode array hardware and software. E.G.W., E.P.S., and E.J.C. wrote the manuscript.

## Competing interests

The authors declare no competing interests.

## Additional information

Eric G. Wu or E. J. Chichilnisky.

**Peer review information** *Nature Communications* thanks Bruno
Olshausen and the other, anonymous, reviewer(s) for their contribution
to the peer review of this work. A peer review file is available.

