## [Peer Review File · Nature Communications]

Fixational Eye Movements Enhance the Precision of Visual Information Transmitted by the Primate RetinaREVIEWER COMMENTS

Reviewer #1 (Remarks to the Author):

Humans and other species continually move their eyes during the acquisition of visual information. Several studies have reported perceptual improvements from these fixational eye movements, and it has been suggested that the resulting stimulus motion contributes to the encoding of visual information in the retina. Wu et al investigates this hypothesis quantitatively by using a Bayesian approach to reconstruct the spatial stimulus from the train of responses in a population of retinal ganglion cells. They report that inversion of the neural code improves with increased fixational motion. I find this article very interesting and the results convincing, I have no major concerns. I list below several comments, primarily intended to improve readability and clarify points in which the text is not fully clear.

Specific comments

There are no line numbers in the document, so I will refer my comments to paragraphs and pages.

Second paragraph, page 2. I would recommend removing the “for the first time” statement from this paragraph. There have been multiple articles arguing that fixational eye movements, spike timing, and correlations are important for the retinal code.

Last paragraph, page 4. The first part of this paragraph seems repetitive. It is already stated in the introduction that posterior probability is estimated as the product of the likelihood of the stimulus and the prior of natural images. It would be better to define variables and symbols in the caption of the figure, rather than in the text.

Last paragraph, page 4. The LNBRC model is introduced as generalization of the standard linear-nonlinear-Poisson model. However, it is unclear why it is important to use this generalization rather than the more common L-NL-P. I understand that a comparison comes later in Figure 1f, but it would be useful to expand and explain the expected advantages of the approach at this stage in the text.

Third paragraph, page 5. What are the four MS-SSIM values? The quality of reconstructions obtained from the four different cell types? If so, there is no way to combine these populations into a single

estimate? These different types of neurons presumably respond to different features and frequencies in the stimulus.

Fourth paragraph, page 5. MS-SSIM values for the full model are already reported in the previous paragraph. It would be useful to (a) provide an intuition of what a 0.5 change in MS-SSIM entails in terms of image quality; and (b) provide some measure of how variable these results are, given that no probability value is given in the comparison. Do the MS-SSIM values critically depend on the specific set of images tested?

Fifth paragraph, page 5. In the definition of fixational jitter it would help to add “ocular” to “drift”, given that this is the term traditionally used in the literature. Also, given that small saccades often occur when fixating on a target, “ocular drift” is commonly defined as the smooth inter-saccadic fixational motion of the eye.

Fifth paragraph, page 5. This paragraph should first make clear that fixational eye movements enhance fine pattern vision (e.g., Rucci et al, 2007) and acuity (Ratnam et al, 2017; Intoy & Rucci 2020). The two hypotheses listed in the text are mechanisms proposed to explain these findings.

Fifth paragraph, page 5. “The visual system may not have precise knowledge of eye position.” Recent evidence actually suggests the opposite, see Zhao et al, Nature Communications, 2023 or Raghunandan et al, Journal of Vision 2008. Note that the model in Zhao et al also explains the previous studies that reached the opposite conclusion.

Second paragraph, page 6. It would have been more reassuring if stimuli were displayed at higher temporal frequencies. I do not follow why the update of the display was so relatively low given that good monitors now reach 480 Hz.

First paragraph, page 10. What was the role of the abrupt onset of the stimulus in the reconstructions of Figure 2? The appearance and disappearance of the stimulus likely triggered strong responses, as also shown in Fig.2C. This contrast step is, however, a laboratory artifact which does not occur during natural viewing. Outside of the laboratory, stimuli are normally brought in by saccades, which provide very different input stimulation than a sudden step, likely attenuating low spatial frequency (e.g., Mostofi et al, Current Biology 2020). Since a step contrast is a very effective stimulus and an extremely informative transient, which is why it is used so extensively in neurophysiological preparations, it may have actually diminished the role of fixational eye movements. In other words, the fixational motion of the eye may be more important during the natural saccade-fixation cycle, when stimuli do not suddenly appear from nowhere. It would be useful to comment on this issue in the discussion.

Fourth paragraph, page 10. Perhaps I am missing something here, but it seems somewhat obvious that the midget cells had more accurate eye position information given the different sampling resolution in the two groups. Also, I imagine the authors are here simply referring to what happens in their model/dataset rather than suggesting that midget cell are responsible for encoding eye movements in the retina. Some rephrasing would help.

Analysis of correlated firing, pages 11-12. I found this section confusing. What are the author's conclusions here, that signal-induced correlation generally help? Or that correlation caused by fixational movements help? In this context, it has been proposed that fixational jitter causes signal correlations by pre-whitening the input effective in driving retinal ganglion cells (Kuang et al, 2012). This operation emphasizes image discontinuities in the patterns of correlated activity.

Reviewer #2 (Remarks to the Author):

The paper demonstrates the beneficial effect of fixational eye movements on visual perception by attempting to reconstruct the best possible image from the recorded spike trains of retinal ganglion cells in response to natural images with and without eye (simulated) movements. The results clearly demonstrate that eye movements yield better image reconstruction, and that the precise timing of spikes and correlations among neurons are important for this.

This paper is a tour de force. I consider it one of the most significant findings regarding retinal function over the past several years. The role of fixational eye movements in vision has long been debated and theorized about - is it a bug or feature in the design of the visual system? This paper convincingly shows it helps vision. The methods employed are computationally sophisticated employing state of the art machine learning techniques, and the data - large population recordings of midget and parasol roc's in response to time-varying natural images - are simply remarkable. And the analyses are extremely thorough. Putting these together makes it an extremely impactful and important paper. It gives us for the first time a window into the end-to-end (from image to rgc) functioning of the retina during natural vision. I believe it will be of great interest to the vision science community and will be highly cited in the years to come. My congratulations to the authors on such a fine paper.

A few comments/questions:

Intro:

The "Bayesian formalism" for image reconstruction was actually first introduced (to my mind) by Rieke et al., 1997 (Spikes), and I think it would be good to reference that.

Results:

Throughout the paper, fixational eye movements are referred to as "jitter." But the main component is actually a drift motion that has a significant momentum component. Jitter implies a fast back and forth, random motion, which I don't think is a good way to characterize retinal drift. I would encourage the authors to think of a different term, or just use "drift" instead which is the more common term used in the eye movement community.

Related: 2D Brownian motion is not the best model of eye drift. Perhaps a reasonable approximation for the present purposes, but I think its worth mentioning that its not ideal.

Note that one important difference between this study and the previous studies of Ratnam et al. and Anderson et al, is the size of the rgc receptive fields with respect to eye movement. The previous studies considered rgc's in the foveal region, where retinal drift moves the stimulus over multiple receptive fields (e.g., Anderson et al., figure 1A). Here by contrast the entire extent of the eye motion lies pretty much within one rgc receptive field, as shown in Figure 2b. That seems to be an important difference worth mentioning in the paper. The finding here is still important, but the benefit doesn't seem to be coming from integrating information across multiple rgc's as a stimulus moves over them. Somehow it seems it's just the movement within the rf that modulates the neuron's response enough to make a difference. Is that right?

The diagrams illustrating the model and the flow of computations in Figures 1 and 2 are very nice.

Reviewer #3 (Remarks to the Author):

The manuscript examines the role of fixational eye movements on the precision of visual information transmitted by the retina. This topic was addressed in many previous works, and the key advance here is that the analysis is performed on recordings from the primate retina, including several hundred retinal ganglion cells (RGCs) which comprise most of the cells responding to a patch of the visual scene. A decoder is trained to reconstruct the image based on spikes, and the consequences of fixational eye movements are examined based on the decoding accuracy.

The study builds on elaborate tools that were developed in previous works (the high density multielectrode array, the LNBRC model, etc.) and enhances them to some extent with additional computational machinery for incorporating a prior for natural images and for reconstructing the trajectory (similar problems were addressed in Anderson et al, Journal of Vision, 2020). The key novelty is in the decoding under motion from actual spike recordings.

The main conclusion is that fixational eye drifts enhance the information transmitted by the retina, which is a useful result. My understanding, though, is that this is done at high eccentricities, where the whole motion spans a distance comparable to the receptive field size of a single RGC. This greatly limits, in my view, the insights that can be gained. The key questions raised in the literature are on foveal vision, where the motion spans many RGCs, the impact on RGC activity is highly nonlinear (viewed as a function of the trajectory), and the complexity of the decoding problem is much higher. It is impossible to extrapolate from the results of the paper to this regime of interest and thus to the fovea, where the fine details of the visual scene are processed (see also point 1 below).

My reading of the paper is that small eye jitters, comparable in magnitude to the receptive field size of a single RGC, increase the fidelity of the information transmitted to the brain by the array of RGCs, and that most likely, this arises from an increase in the firing rate of RGCs. The methodology could potentially be modified to learn something about questions pertaining to foveal vision, by scaling up the magnitude of imposed drifts, but this would require new experiments. Without such a revision, I see the work as technically impressive, but the significance of the results is limited. Therefore, I think that the work will be appropriate for a more specialized journal, following a revision that will address the comments below.

Main comments

1. Much of the interest in fixational drifts and their impact on vision has focused on the fovea. The reason is clear: in this region the range of the motion is large compared to the receptive field size of

an RGC. In the present manuscript, however, the motion is at most comparable with the size of a receptive field (Fig. 2b). Under these conditions, the motion generates a smooth perturbation to the retinal response. Consequently, the inference problem is probably much easier than with larger motion. Results in this work seem to apply only to peripheral vision, and this diminishes the significance of the work: if fixational motion improves peripheral vision, as argued here, but harms the fidelity of the foveal representation, the outcome is completely different than implied by the main conclusion of the paper, especially when considering that the question of scale is completely absent from the presentation. This is a key issue, which should be clear to the reader. The limitations of the analysis (or its potential relevance) should be discussed in the context of scale (and foveal vs peripheral vision) and in relation to previous experimental and theoretical studies.

2. The mechanism responsible for the increase in performance is not elucidated. The paper only speculates that this has to do with an increase in the spike rate, and it does not provide clear data on this, except for mentioning a high Pearson correlation between the jitter and firing rate: but how exactly does the firing rate depend on the motion? Is this a large or small effect relative to the baseline firing rates without jitter?

3. Furthermore, can specific hypotheses be tested more directly in simulations? For example, will the increase in performance be comparable with simulations, in which spikes generated using the LNRBC model are decoded, while adjusting the overall firing rate in accordance with the increase observed with added jitter? Are there ways to test other hypotheses on how fixational motion could improve the precision of the retinal output?

4. Figure 2 shows examples of reconstructions, where differences in the MS-SSIM score are of order ~ 0.6 . These map to noticeable differences in the quality of reconstruction. Later in the manuscript, reported MS-SSIM differences associated with exclusion of spike correlations (page 12) are much smaller, of order 0.02 to 0.04. It's not surprising that the model with correlations, which includes more parameters performs a bit better than the model that excludes correlations. However, should we interpret a difference of 0.04 as indicating that correlations play an important role? Naively, at least, the small differences may indicate the opposite. They certainly do not seem to suggest that knowledge of correlated firing properties is necessary to effectively decode image content – the conclusion made in page 12, or that correlated RGC firing is critical for reconstructing jittered natural images as stated in Results.

5. Statistics of motion are modeled as Brownian motion, but examination of these statistics, both in humans and recently in non-human primates indicates that this is a poor description. The motion is more persistent than Brownian motion (Chrici et al, 2012; Hermann et al, 2017; Ben-Shushan et al, 2022).

6. The presentation is often obscure due to a vague choice of quantitative measures, units, etc. For example – there is no scalebar in Fig. 2b. The axes in Figure 4 are incomprehensible (see point 10 below). The qualitative meaning of the MS-SSIM measure is not explained (is a difference in values of 0.02 large or small?). The results section does not state what is the patch size from which measurements are available, what is the eccentricity, what is the typical size of a receptive field in this region and how all this relates to the magnitude of fixational motion. The amount of motion, and how it is quantified, is vaguely and inconsistently described (see comments 8 and 9).

More minor comments

7. The Introduction states that the importance of eye movements is demonstrated here “for the first time”. In my opinion, this statement is incorrect without qualifications, since the importance of eye movements was demonstrated in previous psychophysical studies and theoretical works.

8. In page 6 the motion is described as having a standard deviation of 10 micrometers per frame. The methods section mentions a diffusion coefficient of 10 micrometers squared per frame. These two descriptions are inconsistent.

9. Later on, results are presented as a function of the amount of jitter. Was the diffusion coefficient modified? Or do the quantities ‘Eye position std’ and ‘eye jitter magnitude’ represent a standard deviation within a single trial? How exactly is this defined? Do the terms ‘Eye position std’ and ‘eye jitter magnitude’ represent the same quantity?

10. In Figure 4, what type of scale is used in the horizontal axis? The location of the tick marks labeled (0, 10^0 , 10^1) don’t match a linear scale, but they cannot match a logarithmic scale either, since 0 should not appear on such a scale. If the scale is logarithmic (and the label ‘0’ is not correct), I am not convinced that reconstruction quality declines slowly and then more sharply (top of page 11).

11. In the inference algorithm for the trajectory distribution q , the underlying model is Markovian. This implies that marginals of the distribution q at specific time points can be evaluated precisely using a Markov decoding algorithm. It would be informative to compare this precise calculation with the marginals obtained from the particle filter approach, to assess the precision of the algorithm.

12. Note the typo on Page 8: “exceeded was better than”.

13. References are difficult to identify, because they are cited using the first author's name and year of publication, whereas the numbered reference list is not ordered based on these keys.

Reviewer #4 (Remarks to the Author):

This manuscript addresses a longstanding and important question about the role of fixational drift in vision. It has been known for over 100 years that the retina is never still, and many have conjectured that these micro-eye-movements play an important role in the visual encoding process. In the last decade, psychophysical evidence has converged that the visual system utilizes drift in the process of seeing (especially at the center of gaze). However, to date, no physiological studies have demonstrated whether information is actually increased by fixational jitter. This paper takes a major step in answering this question and it does so using an impressive combination of population recordings in the ex-vivo retina and state-of-the art image-reconstruction algorithms. This manuscript provides compelling evidence that the retina is sensitive to realistic amounts of jitter and that such jitter increases information. The approach is innovative, the methods are both sound and impressive, and the writing is clear. Overall, this manuscript sets a high bar for the

I have no major comments, but I have several minor comments that I think would improve the quality and impact of the manuscript. Some of these are more minor than others so I've put the most important one at the top.

1) The eccentricity of these recordings is missing from the methods and is an important detail that should be discussed. My best guess, given previous recordings from the Chichilnisky lab, is these are fairly peripheral recordings. This makes the result that fixational drift adds information even more impressive, but also suggests this is a lower bound on the role these eye movements play. Virtually all psychophysical evidence has investigated drift near the center of gaze. At a minimum, this manuscript should discuss this point. If the authors are feeling more ambitious, I think it would be helpful to relate the jitter in their recordings to the spacing and size of cones at this eccentricity. This would be helpful in contextualizing these results with respect to the psychophysical studies and, I think, plant an important seed for future work.

2) In the abstract, the authors suggest they present evidence for more precise temporal coding than has been suggested by previous studies. This point is revisited in the discussion, but one recent study should likely be included here:

Liu B, Hong A, Rieke F, Manookin MB. Predictive encoding of motion begins in the primate retina. *Nature neuroscience*. 2021 Sep;24(9):1280-91.

Liu et al., 2021 demonstrates that information about the motion of a stimulus depends on the timing of spikes at roughly 1ms precision. They show that jitter in spike times on the order of 1ms reduces information about motion and that is consistent with the results here if I'm not thinking about this incorrectly.

3) This is minor, but in the section "Bayesian reconstruction of images displayed with fixational eye movements", the authors suggest "that the visual system may not have precise knowledge of the eye position". I agree that is a good starting point and the joint estimation procedure is compelling evidence the brain doesn't need an extra-retinal signal to benefit from fixational jitter, but recent evidence from Michele Rucci and Jonathan Victor suggests it might have such a signal:

Zhao Z, Ahissar E, Victor JD, Rucci M. Inferring visual space from ultra-fine extra-retinal knowledge of gaze position. *Nature communications*. 2023 Jan 17;14(1):269.

For completeness, this should likely show up as a "but see" paper in this section, because your results explore both options: eye position known and inferred...

4) The LNBRC model is noticeably less sparse than the data, which appear qualitatively more "episodic" in their responses. The authors highlight the benefits of having a convex model architecture for actually fitting something to data, but would a more biophysically accurate model have the potential to change the size of this effect? Naively, one might think that a more precise model would support an even bigger effect of eye movements. The discussion mentions recent deep neural network / subunit approaches, but the CNN approaches have not operated at 1ms time resolution. These results suggest that temporally precise models will be necessary for accurately assessing the information in spike trains from the retina.

5) Minor: I think the citation is missing for the FISTA algorithm in the methods.

Fovea overall response

The key question/concern raised by all four reviewers is that the magnitudes of the (simulated) naturalistic drift eye movements tested in our experiments are smaller than the receptive field sizes of RGCs in the peripheral retina, and thus the results do not necessarily generalize to retinal coding in the fovea, where typical eye movements are larger than RGC receptive fields.

We agree that the manuscript has this limitation, because large-scale recordings from complete RGC populations in the fovea are not currently feasible. But we do not believe that this makes the work less interesting or diminishes its overall significance. The paper develops a novel computational framework to simultaneously decode complex natural scenes and motion associated with drift eye movements directly from recorded spikes in complete populations of RGCs, and uses this framework to demonstrate that drift eye movements substantially improve the fidelity of the retina code. This improvement, which has been suggested and discussed in many previous studies, is central to our understanding of natural vision, but has never before been directly quantified. Although our paper analyzed only peripheral retina, our computational approach is general, and will be applicable to large-scale recordings in the fovea when this becomes technologically feasible. In addition, we disagree with the assertion of reviewer 3 that foveal decoding in the presence of drift eye movements will be substantially more difficult or more interesting than in the periphery (reviewers 2 and 4 clearly agree with us on this point). Our intuition is that drift compensation should be, if anything, easier in the fovea, because estimates of image displacement are easier with smaller, denser sampling of the image. If so, the demonstrated enhancement of retinal signals by drift is more surprising in the peripheral retina, where the receptive fields are larger and more sparsely sampled.

However, we completely agree with the reviewers that we should have addressed this issue more explicitly in our manuscript. To this end, we have modified the manuscript as follows:

1. We have much more clearly and directly described the magnitudes of simulated eye movements tested in relation to the RGC receptive field sizes and spacing in our recordings in a new paragraph of the Results connected to Fig. 3. We also provide a simple analysis demonstrating that the quality of reconstructed images improves even though the magnitude of realized eye movements sometimes exceeds the radius of a single midget cell receptive field. Although these findings cannot be directly extrapolated to the fovea, where naturalistic eye movements traverse multiple receptive fields, they do suggest that the improvements in image quality associated with increasing eye movements continue to hold when the scale of motion is larger than one receptive field. However, we have also made clear that the bulk of the data in the manuscript is obtained with smaller image displacements.

2. To additionally facilitate comparison of the magnitudes of eye movements with receptive field size, we have added scale bars to Figure 2b. We have also described the eccentricities of the recordings in the Methods.
3. We have added a paragraph to the Discussion about the relationship between our results and retinal coding in the fovea. This paragraph briefly describes how the differences in receptive field size and cell density between fovea and periphery may affect retinal coding in the presence of eye movements, the fact that our results may or may not translate to the fovea, and the fact that our techniques and analysis will be applicable to foveal RGC population recordings when such experiments become possible.

Below we address the other comments by the reviewers point-by-point.

Reviewer 1

Humans and other species continually move their eyes during the acquisition of visual information. Several studies have reported perceptual improvements from these fixational eye movements, and it has been suggested that the resulting stimulus motion contributes to the encoding of visual information in the retina. Wu et al investigates this hypothesis quantitatively by using a Bayesian approach to reconstruct the spatial stimulus from the train of responses in a population of retinal ganglion cells. They report that inversion of the neural code improves with increased fixational motion. I find this article very interesting and the results convincing, I have no major concerns. I list below several comments, primarily intended to improve readability and clarify points in which the text is not fully clear.

Thank you for these constructive comments. Detailed responses are provided below.

Specific comments

There are no line numbers in the document, so I will refer my comments to paragraphs and pages.

Second paragraph, page 2. I would recommend removing the “for the first time” statement from this paragraph. There have been multiple articles arguing that fixational eye movements, spike timing, and correlations are important for the retinal code.

We agree, and have removed the phrase.

Last paragraph, page 4. The first part of this paragraph seems repetitive. It is already stated in the introduction that posterior probability is estimated as the product of the likelihood of the stimulus and the prior of natural images. It would be better to define variables and symbols in the caption of the figure, rather than in the text.

Last paragraph, page 4. The LNBRC model is introduced as generalization of the standard linear-nonlinear-Poisson model. However, it is unclear why it is important to use this generalization rather than the more common L-NL-P. I understand that a comparison comes later in Figure 1f, but it would be useful to expand and explain the expected advantages of the approach at this stage in the text.

Thanks for this suggestion: we have added a sentence in that paragraph explaining the expected advantages of the LNBRC over the simpler LNP alternative.

Third paragraph, page 5. What are the four MS-SSIM values? The quality of reconstructions obtained from the four different cell types? If so, there is no way

to combine these populations into a single estimate? These different types of neurons presumably respond to different features and frequencies in the stimulus.

The four MS-SSIM values on page 5 are computed for each of four distinct experimental preparations from four animals. These are averaged over reconstructions computed using all of the cells (ON and OFF parasol and midget cells) in each respective recording. We have changed the language to explicitly say this wherever a sequence of values is listed.

A brief cell-type specific analysis is presented in Figure 3cd.

Fourth paragraph, page 5. MS-SSIM values for the full model are already reported in the previous paragraph. It would be useful to (a) provide an intuition of what a 0.5 change in MS-SSIM entails in terms of image quality; and (b) provide some measure of how variable these results are, given that no probability value is given in the comparison. Do the MS-SSIM values critically depend on the specific set of images tested?

MS-SSIM provides an approximate measure of perceptual quality that is well-aligned with human opinion scores over a wide range of different types of distortion. Nevertheless, it is difficult to give a straightforward qualitative intuition about the meaning of given change in MS-SSIM beyond showing representative example images (e.g. Figure 1f for flashed reconstructions, Supplemental Figure S2 for eye movements reconstructions). The images tested for each reconstruction approach are the same, so all model comparisons are “apples to apples”.

We have now documented the (highly significant) differences in the performance of the full LNBRC-dCNN method relative to the simpler encoding model using non-parametric paired order tests.

Fifth paragraph, page 5. In the definition of fixational jitter it would help to add “ocular” to “drift”, given that this is the term traditionally used in the literature. Also, given that small saccades often occur when fixating on a target, “ocular drift” is commonly defined as the smooth inter-saccadic fixational motion of the eye.

Thanks for this suggestion. We have changed the phrasing to “ocular drift” in the definition, and have replaced the term “jitter” with “fixational drift” everywhere in the manuscript.

Fifth paragraph, page 5. This paragraph should first make clear that fixational eye movements enhance fine pattern vision (e.g., Rucci et al, 2007) and acuity (Ratnam et al, 2017; Intoy & Rucci 2020). The two hypotheses listed in the text are mechanisms proposed to explain these findings.

We have modified the corresponding paragraph to this effect.

Fifth paragraph, page 5. “The visual system may not have precise knowledge of eye position.” Recent evidence actually suggests the opposite, see Zhao et al, Nature Communications, 2023 or Raghunandan et al, Journal of Vision 2008. Note that the model in Zhao et al also explains the previous studies that reached the opposite conclusion.

Thank you for suggesting these additional references. We’ve included them at the end of the sentence where we acknowledge the unresolved nature of this issue. We have also added citations in the paragraph introducing fixational drift.

Second paragraph, page 6. It would have been more reassuring if stimuli were displayed at higher temporal frequencies. I do not follow why the update of the display was so relatively low given that good monitors now reach 480 Hz.

The 120 Hz monitors we used for these experiments are sufficient to exceed the temporal resolution of the parasol cells. Specifically, the autocorrelation of parasol cell spike trains do not exhibit 120 Hz oscillatory structure, as would be expected if the cells reliably resolved this frequency. Indeed, parasol cells do sometimes show oscillatory autocorrelation with lower refresh rates (e.g. 60 Hz). Given that parasol cells are thought to provide the primary signals about image transients to the brain, this refresh frequency seems adequate for the present analysis. Note that a new visual stimulation apparatus with faster refresh would require substantial additional data collection, a task that would likely take 1-2 years to complete.

First paragraph, page 10. What was the role of the abrupt onset of the stimulus in the reconstructions of Figure 2? The appearance and disappearance of the stimulus likely triggered strong responses, as also shown in Fig.2C. This contrast step is, however, a laboratory artifact which does not occur during natural viewing. Outside of the laboratory, stimuli are normally brought in by saccades, which provide very different input stimulation than a sudden step, likely attenuating low spatial frequency (e.g., Mostofi et al, Current Biology 2020). Since a step contrast is a very effective stimulus and an extremely informative transient, which is why it is used so extensively in neurophysiological preparations, it may have actually diminished the role of fixational eye movements. In other words, the fixational motion of the eye may be more important during the natural saccade-fixation cycle, when stimuli do not suddenly appear from nowhere. It would be useful to comment on this issue in the discussion.

We agree that the transition between consecutive stimuli may produce a stronger transient response in the RGCs than that which may occur with true saccadic eye movements, and that this transient response may contribute substantially to the reconstructions and thus diminish the observed effect size for the drift eye movement magnitude in our results in comparison to natural vision. We have added a brief

paragraph to the Discussion, including this point, about the potential impacts of stimulus design on the results.

Fourth paragraph, page 10. Perhaps I am missing something here, but it seems somewhat obvious that the midget cells had more accurate eye position information given the different sampling resolution in the two groups. Also, I imagine the authors are here simply referring to what happens in their model/dataset rather than suggesting that midget cell are responsible for encoding eye movements in the retina. Some rephrasing would help.

We think that the comparison of position information between parasol and midget RGCs is not entirely obvious: while the midget cells do have substantially smaller and denser receptive fields, they temporally integrate the visual stimulus more slowly than parasol cells, and this temporal blur could degrade their encoding of eye position. Our analysis shows that this is not the case. We have added a sentence to introduce the analysis so that the reader is alerted that the outcome is not obvious.

Analysis of correlated firing, pages 11-12. I found this section confusing. What are the author's conclusions here, that signal-induced correlation generally help? Or that correlation caused by fixational movements help? In this context, it has been proposed that fixational jitter causes signal correlations by pre-whitening the input effective in driving retinal ganglion cells (Kuang et al, 2012). This operation emphasizes image discontinuities in the patterns of correlated activity.

The main finding of this section is that exploiting signal-induced correlations, which are partly captured by the coupling filters of the LNBRC, improves image reconstruction from RGC responses. To be clear, we are *not* attempting to make an explicit comment about correlated firing induced specifically by eye movements – this is difficult to tease out in the present data.

The correlations section was also intended to clarify and correct common misconceptions in the retinal encoding model literature about how generalized linear models (GLMs, which include the LNBRC) represent RGC responses to natural scenes. One common misconception is that the neighboring cell coupling filters in the GLM exclusively serve to capture noise-correlated firing in RGCs. While this is the intent of the GLM architecture, and indeed is true in the case of uncorrelated visual stimuli [e.g. Pillow *et al.*, ref. 1 in the main text], our analysis shows that in the case of naturalistic stimuli, the coupling filters largely instead capture stimulus-induced correlations. This interpretation of the coupling filters has led to a second common misconception about the GLM. A highly-cited preprint by our group [1, below] used *uncoupled* GLMs to suggest that the GLM cannot accurately describe retinal responses to natural scenes. The results of this section demonstrate that this statement does not apply to coupled GLMs, and that the inclusion of the coupling filters (representing stimulus-induced correlations) in particular is responsible for the discrepancy.

Based on the reviewer's questions, we have reworded some of this section in order to ensure proper emphasis (though we have not attempted to explain the last two points above in detail because it seems like the wrong emphasis for this paper).

[1] Heitman, A. *et al.* *Testing Pseudo-Linear Models of Responses to Natural Scenes in Primate Retina*. <http://biorxiv.org/lookup/doi/10.1101/045336> (2016)
doi:[10.1101/045336](https://doi.org/10.1101/045336).

Reviewer 2

The paper demonstrates the beneficial effect of fixational eye movements on visual perception by attempting to reconstruct the best possible image from the recorded spike trains of retinal ganglion cells in response to natural images with and without eye (simulated) movements. The results clearly demonstrate that eye movements yield better image reconstruction, and that the precise timing of spikes and correlations among neurons are important for this.

This paper is a tour de force. I consider it one of the most significant findings regarding retinal function over the past several years. The role of fixational eye movements in vision has long been debated and theorized about - is it a bug or feature in the design of the visual system? This paper convincingly shows it helps vision. The methods employed are computationally sophisticated employing state of the art machine learning techniques, and the data - large population recordings of midget and parasol roc's in response to time-varying natural images - are simply remarkable. And the analyses are extremely thorough. Putting these together makes it an extremely impactful and important paper. It gives us for the first time a window into the end-to-end (from image to rgc) functioning of the retina during natural vision. I believe it will be of great interest to the vision science community and will be highly cited in the years to come. My congratulations to the authors on such a fine paper.

Thank you for these constructive comments. Detailed responses are provided below.

A few comments/questions:

Intro:

The "Bayesian formalism" for image reconstruction was actually first introduced (to my mind) by Rieke et al., 1997 (*Spikes*), and I think it would be good to reference that.

We've added a citation to *Spikes* to the introduction.

Results:

Throughout the paper, fixational eye movements are referred to as "jitter." But the main component is actually a drift motion that has a significant momentum component. Jitter implies a fast back and forth, random motion, which I don't think is a good way to characterize retinal drift. I would encourage the authors to think of a different term, or just use "drift" instead which is the more common term used in the eye movement community.

We agree with this (and with a related comment by Rev 1). We have changed the phrasing to “ocular drift” in the definition, and have replaced the term “jitter” with “fixational drift” everywhere in the manuscript.

Related: 2D Brownian motion is not the best model of eye drift. Perhaps a reasonable approximation for the present purposes, but I think its worth mentioning that its not ideal.

We agree that Brownian motion is an imperfect description of drift eye movements. However, since the decoding algorithm makes minimal assumptions about the statistical structure of the stimulus, we believe that the specific choice of Brownian motion is unlikely to significantly affect the results. Moreover, Brownian motion is maximally entropic and hence more random and more difficult to predict than real drift eye movements. Thus, our estimate of reconstruction quality is likely to be worse (e.g. a lower bound) than if a more realistic model for drift was used. We have added a paragraph to the Discussion addressing possible impacts of stimulus design choices on the results.

Note that one important difference between this study and the previous studies of Ratnam et al. and Anderson et al, is the size of the rgc receptive fields with respect to eye movement. The previous studies considered rgc's in the foveal region, where retinal drift moves the stimulus over multiple receptive fields (e.g., Anderson et al., figure 1A). Here by contrast the entire extent of the eye motion lies pretty much within one rgc receptive field, as shown in Figure 2b. That seems to be an important difference worth mentioning in the paper. The finding here is still important, but the benefit doesn't seem to be coming from integrating information across multiple rgc's as a stimulus moves over them. Somehow it seems it's just the movement within the rf that modulates the neuron's response enough to make a difference. Is that right?

See response in the introductory portion of this document.

The diagrams illustrating the model and the flow of computations in Figures 1 and 2 are very nice.

Thanks for your positive feedback!

Reviewer 3

The manuscript examines the role of fixational eye movements on the precision of visual information transmitted by the retina. This topic was addressed in many previous works, and the key advance here is that the analysis is performed on recordings from the primate retina, including several hundred retinal ganglion cells (RGCs) which comprise most of the cells responding to a patch of the visual scene. A decoder is trained to reconstruct the image based on spikes, and the consequences of fixational eye movements are examined based on the decoding accuracy.

The study builds on elaborate tools that were developed in previous works (the high density multielectrode array, the LNBRC model, etc.) and enhances them to some extent with additional computational machinery for incorporating a prior for natural images and for reconstructing the trajectory (similar problems were addressed in Anderson et al, Journal of Vision, 2020). The key novelty is in the decoding under motion from actual spike recordings.

The main conclusion is that fixational eye drifts enhance the information transmitted by the retina, which is a useful result. My understanding, though, is that this is done at high eccentricities, where the whole motion spans a distance comparable to the receptive field size of a single RGC. This greatly limits, in my view, the insights that can be gained. The key questions raised in the literature are on foveal vision, where the motion spans many RGCs, the impact on RGC activity is highly nonlinear (viewed as a function of the trajectory), and the complexity of the decoding problem is much higher. It is impossible to extrapolate from the results of the paper to this regime of interest and thus to the fovea, where the fine details of the visual scene are processed (see also point 1 below).

My reading of the paper is that small eye jitters, comparable in magnitude to the receptive field size of a single RGC, increase the fidelity of the information transmitted to the brain by the array of RGCs, and that most likely, this arises from an increase in the firing rate of RGCs. The methodology could potentially be modified to learn something about questions pertaining to foveal vision, by scaling up the magnitude of imposed drifts, but this would require new experiments. Without such a revision, I see the work as technically impressive, but the significance of the results is limited. Therefore, I think that the work will be appropriate for a more specialized journal, following a revision that will address the comments below.

Thank you for these constructive comments. Detailed responses are provided below.

Main comments

1. Much of the interest in fixational drifts and their impact on vision has focused on the fovea. The reason is clear: in this region the range of the motion is large compared to the receptive field size of an RGC. In the present manuscript, however, the motion is at most comparable with the size of a receptive field (Fig. 2b). Under these conditions, the motion generates a smooth perturbation to the retinal response. Consequently, the inference problem is probably much easier than with larger motion. Results in this work seem to apply only to peripheral vision, and this diminishes the significance of the work: if fixational motion improves peripheral vision, as argued here, but harms the fidelity of the foveal representation, the outcome is completely different than implied by the main conclusion of the paper, especially when considering that the question of scale is completely absent from the presentation. This is a key issue, which should be clear to the reader. The limitations of the analysis (or its potential relevance) should be discussed in the context of scale (and foveal vs peripheral vision) and in relation to previous experimental and theoretical studies.

We have addressed this main comment, and similar comments by the other reviewers, collectively in the introductory portion of this document.

We additionally note that Reviewer 4 directly disagrees with the idea that the inference problem is easier with smaller motion, as do we.

2. The mechanism responsible for the increase in performance is not elucidated. The paper only speculates that this has to do with an increase in the spike rate, and it does not provide clear data on this, except for mentioning a high Pearson correlation between the jitter and firing rate: but how exactly does the firing rate depend on the motion? Is this a large or small effect relative to the baseline firing rates without jitter?

We agree that a clearer presentation of the relationship between eye movement magnitude and spike rate would be useful. We have added a sentence with summary statistics describing the mean rate of change in spike rate as a function of eye movement magnitude, and we have clarified a sentence in the main text to precisely describe the correlation between eye movement magnitude and firing rate. While our results show that increased eye movements are associated with higher firing rates and improved reconstructed image quality, we were unable to show that the relationship between firing rate and image quality is causal, or to quantify the effect size, and thus we can only speculate about the association between image quality and firing rate. The revised text attempts to capture this.

3. Furthermore, can specific hypotheses be tested more directly in simulations? For example, will the increase in performance be comparable with simulations, in which spikes generated using the LNRBC model are decoded, while adjusting the overall firing rate in accordance with the increase observed with added jitter? Are there ways to test other hypotheses on how fixational motion could improve the precision of the retinal output?

Although we agree that in principle simulations could be helpful, and have explored this possibility, it turns out that their interpretation is more complex than one might initially imagine. For example, while the fitted encoding models explain a large fraction of the variance in the observed firing rates of individual RGCs (Supplemental Figure S1), they do have systematic biases, and simulations of full recorded populations of several hundred cells (rather than individual cells) sometimes produce unstable results that differ substantially from real recorded responses. Because of these limitations, we think that including simulations, with all the necessary caveats and controls for such issues, is beyond the scope of the paper.

4. Figure 2 shows examples of reconstructions, where differences in the MS-SSIM score are of order ~ 0.6 . These map to noticeable differences in the quality of reconstruction. Later in the manuscript, reported MS-SSIM differences associated with exclusion of spike correlations (page 12) are much smaller, of order 0.02 to 0.04. It's not surprising that the model with correlations, which includes more parameters performs a bit better than the model that excludes correlations. However, should we interpret a difference of 0.04 as indicating that correlations play an important role? Naively, at least, the small differences may indicate the opposite. They certainly do not seem to suggest that knowledge of correlated firing properties is necessary to effectively decode image content – the conclusion made in page 12, or that correlated RGC firing is critical for reconstructing jittered natural images as stated in Results.

We believe that the reviewer is mistaken in their interpretation of Figure 2, and their comparison between Figure 2g and the correlations analysis is incorrect. Figure 2g plots *relative MS-SSIM* of joint-LNBRC-dCNN - the *relative* image quality of the joint reconstructions in comparison to the reconstructions using zero and exact eye movements, and is computed by mapping the MS-SSIM for zero-LNBRC-dCNN to 0 (e.g. every images in the third column of Figure 2e is mapped to 0 via subtraction), mapping known-LNBRC-dCNN to 1 (e.g. every image in the second column of Figure 2e is mapped to 1 via subtraction and division), and then determining where on this 0-1 scale each of the joint reconstructions fall. This is denoted on Figure 2g by the dotted and solid horizontal lines, and 0 and 1 are labeled in the panel accordingly. In contrast, the correlations analysis is performed using raw MS-SSIM, and cannot be compared with Figure 2g. To avoid this confusion, we have modified the y-axis label in Figure 2g to say “relative quality” rather than relative MS-SSIM, and have modified the corresponding figure caption to more clearly describe the normalization procedure.

5. Statistics of motion are modeled as Brownian motion, but examination of these statistics, both in humans and recently in non-human primates indicates that this is a poor description. The motion is more persistent than Brownian motion (Chrici et al, 2012; Hermann et al, 2017; Ben-Shushan et al, 2022).

We agree that Brownian motion is an imperfect description of drift eye movements. However, because our reconstruction algorithm makes fairly minimal assumptions about

the structure of eye movements, we do not think that the use of Brownian motion for eye movements significantly biases the results of the study. Moreover, Brownian motion is maximally entropic and hence more random than real drift eye movements. As such, compensation for unknown Brownian motion is more difficult than for real drift eye movements, and thus our estimate of reconstruction quality with eye movements is likely to be worse (e.g. a lower bound) than if a more realistic model for drift was used. Finally, we think that Brownian motion is a reasonable simplifying assumption – designing highly-realistic eye movements stimuli accounting for all properties of eye movements for *ex vivo* recordings would be difficult.

We have added a paragraph to the Discussion addressing the possible impacts of the choices made in designing the stimulus. The paragraph mentions this as a possible limitation in the stimulus design.

6. The presentation is often obscure due to a vague choice of quantitative measures, units, etc. For example – there is no scalebar in Fig. 2b. The axes in Figure 4 are incomprehensible (see point 10 below). The qualitative meaning of the MS-SSIM measure is not explained (is a difference in values of 0.02 large or small?). The results section does not state what is the patch size from which measurements are available, what is the eccentricity, what is the typical size of a receptive field in this region and how all this relates to the magnitude of fixational motion. The amount of motion, and how it is quantified, is vaguely and inconsistently described (see comments 8 and 9).

More minor comments

7. The Introduction states that the importance of eye movements is demonstrated here “for the first time”. In my opinion, this statement is incorrect without qualifications, since the importance of eye movements was demonstrated in previous psychophysical studies and theoretical works.

We agree that the phrasing could be misinterpreted and we regret this oversight. What we wrote was that the importance of eye movements “for the retinal code” was demonstrated for the first time – which is accurate – but the reviewer is certainly correct that the importance for behavioral visual performance has already been shown. We have removed “for the first time” to avoid giving the wrong impression.

8. In page 6 the motion is described as having a standard deviation of 10 micrometers per frame. The methods section mentions a diffusion coefficient of 10 micrometers squared per frame. These two descriptions are inconsistent.

Thanks: The correct value of the diffusion constant is 10 micrometers squared per frame, a spatial variance measure appropriate for random Gaussian draws – we have corrected this in the text. For clarity, please note that the figures use instead a standard deviation measure of the entire *realized* eye movement trajectories (rather than per-

frame variance). The units of this measure (micrometers) makes it possible to compare the spatial scale of the full trajectory to the spatial scale of receptive fields and images.

9. Later on, results are presented as a function of the amount of jitter. Was the diffusion coefficient modified? Or do the quantities ‘Eye position std’ and ‘eye jitter magnitude’ represent a standard deviation within a single trial? How exactly is this defined? Do the terms ‘Eye position std’ and ‘eye jitter magnitude’ represent the same quantity?

The diffusion coefficient is not modified. The generation of the Brownian motion trajectories is a random process, and ‘Eye position std’ refers to the *realized* standard deviation occurring within a single stimulus presentation – in other words, the standard deviation in eye position that came about in a particular trial as a result of the stochastic process. We do agree that the wording is confusing, and have added the term “realized” throughout and removed the term ‘Eye position std’ everywhere.

10. In Figure 4, what type of scale is used in the horizontal axis? The location of the tick marks labeled (0, 10^0 , 10^1) don’t match a linear scale, but they cannot match a logarithmic scale either, since 0 should not appear on such a scale. If the scale is logarithmic (and the label ‘0’ is not correct), I am not convinced that reconstruction quality declines slowly and then more sharply (top of page 11).

We have replaced the x-axis with a broken x-axis, where the datapoints $\geq 10^0$ are plotted on a logarithmic scale, while the datapoints for 0 are plotted separately.. We have added a brief comment on this in the caption for clarity.

11. In the inference algorithm for the trajectory distribution q , the underlying model is Markovian. This implies that marginals of the distribution q at specific time points can be evaluated precisely using a Markov decoding algorithm. It would be informative to compare this precise calculation with the marginals obtained from the particle filter approach, to assess the precision of the algorithm.

We would appreciate further clarification on this comment. The definition of q contains a likelihood factor $p(\mathbf{s} | \mathbf{y}, \mathbf{w}_{0:T})$ corresponding to the LNBRC encoding model, which has an explicit temporal history dependence on $\mathbf{w}_{0:T}$, representing the total eye movement trajectory (the eye position history over all timesteps). This dependence results from the stimulus temporal integration properties of the encoding model. Exact evaluation of $q(\mathbf{w})$ would require evaluation of this likelihood factor over all possible eye movement trajectories, which has exponential complexity in the number of timesteps considered and is computationally intractable for the temporal integration windows used here.

Furthermore, because the likelihood factor in q explicitly depends on the estimate of the reconstructed image \mathbf{y} as well, this analysis may also require making significant assumptions about \mathbf{y} , which may substantially alter the interpretation of any lower bound derived in this way.

If the reviewer would like to follow up with this, we would be glad to discuss further.

12. Note the typo on Page 8: “exceeded was better than”.

Thanks, we have fixed the error.

13. References are difficult to identify, because they are cited using the first author’s name and year of publication, whereas the numbered reference list is not ordered based on these keys.

We have added references that were erroneously missing from the manuscript, deleted other references that are no longer cited by the manuscript, and re-ordered the references to be in order of appearance.

Reviewer 4

This manuscript addresses a longstanding and important question about the role of fixational drift in vision. It has been known for over 100 years that the retina is never still, and many have conjectured that these micro-eye-movements play an important role in the visual encoding process. In the last decade, psychophysical evidence has converged that the visual system utilizes drift in the process of seeing (especially at the center of gaze). However, to date, no physiological studies have demonstrated whether information is actually increased by fixational jitter. This paper takes a major step in answering this question and it does so using an impressive combination of population recordings in the ex-vivo retina and state-of-the art image-reconstruction algorithms. This manuscript provides compelling evidence that the retina is sensitive to realistic amounts of jitter and that such jitter increases information. The approach is innovative, the methods are both sound and impressive, and the writing is clear. Overall, this manuscript sets a high bar for the

I have no major comments, but I have several minor comments that I think would improve the quality and impact of the manuscript. Some of these are more minor than others so I've put the most important one at the top.

1) The eccentricity of these recordings is missing from the methods and is an important detail that should be discussed. My best guess, given previous recordings from the Chichilnisky lab, is these are fairly peripheral recordings. This makes the result that fixational drift adds information even more impressive, but also suggests this is a lower bound on the role these eye movements play. Virtually all psychophysical evidence has investigated drift near the center of gaze. At a minimum, this manuscript should discuss this point.

We have addressed this important comment, and similar comments by the other reviewers, collectively in the introductory portion of this document.

If the authors are feeling more ambitious, I think it would be helpful to relate the jitter in their recordings to the spacing and size of cones at this eccentricity. This would be helpful in contextualizing these results with respect to the psychophysical studies and, I think, plant an important seed for future work.

We estimate that the mean spacing between cones in the preparations used is around 14 microns. Needless to say, the fixational drift we use is larger than the typical cone spacing, as we now report in the text related to Fig. 3.

2) In the abstract, the authors suggest they present evidence for more precise temporal coding than has been suggested by previous studies. This point is revisited in the discussion, but one recent study should likely be included here:

Liu B, Hong A, Rieke F, Manookin MB. Predictive encoding of motion begins in the primate retina. Nature neuroscience. 2021 Sep;24(9):1280-91.

Liu et al., 2021 demonstrates that information about the motion of a stimulus depends on the timing of spikes at roughly 1ms precision. They show that jitter in spike times on the order of 1ms reduces information about motion and that is consistent with the results here if I'm not thinking about this incorrectly.

We thank the reviewer for their comment. This is an excellent suggestion, and we have added a citation to this work in the Discussion.

3) This is minor, but in the section “Bayesian reconstruction of images displayed with fixational eye movements”, the authors suggest “that the visual system may not have precise knowledge of the eye position”. I agree that is a good starting point and the joint estimation procedure is compelling evidence the brain doesn't need an extra-retinal signal to benefit from fixational jitter, but recent evidence from Michele Rucci and Jonathan Victor suggests it might have such a signal:

Zhao Z, Ahissar E, Victor JD, Rucci M. Inferring visual space from ultra-fine extra-retinal knowledge of gaze position. Nature communications. 2023 Jan 17;14(1):269.

For completeness, this should likely show up as a "but see" paper in this section, because your results explore both options: eye position known and inferred...

We thank the reviewer for this comment, and we have added a citation to this work in the paragraph introducing the eye movements stimulus. We have also put our results in context of this work in the Discussion.

4) The LNBRC model is noticeably less sparse than the data, which appear qualitatively more “episodic” in their responses. The authors highlight the benefits of having a convex model architecture for actually fitting something to data, but would a more biophysically accurate model have the potential to change the size of this effect? Naively, one might think that a more precise model would support an even bigger effect of eye movements. The discussion mentions recent deep neural network / subunit approaches, but the CNN approaches have not operated at 1ms time resolution. These results suggest that temporally precise models will be necessary for accurately assessing the information in spike trains from the retina.

We thank the reviewer for their comment. A more biophysically accurate model that improves the accuracy of RGC response modeling relative to the LNBRC could in principle change the size of the effect. Because such a model would more accurately

capture retinal responses to modulations in luminance associated with eye movements, our intuition matches that of the reviewer, that the effects of eye movements would be greater. This comes with a major caveat: encoding models including additional complexity would need to be paired with an appropriate reconstruction method to maximally benefit from the improved encoding model accuracy. The benefit of the LNBRC model is that it allows for convex optimization of likelihood, which in turn makes the optimal Bayesian estimation process tractable. We don't know how we would use a more biophysically interpretable model (e.g. <https://pubmed.ncbi.nlm.nih.gov/11430813/> or <https://pubmed.ncbi.nlm.nih.gov/16306413/>) to perform a similar Bayesian estimate. This could be an interesting avenue for future research.

We agree with the reviewer that our results suggest that high temporal resolution in an encoding model may be necessary to accurately assess the information contained in retinal spike trains, and that this may limit the reconstruction quality produced using current-day CNN retinal encoding models. We have modified the discussion of temporal precision in the third-to-last paragraph of the Discussion.

5) Minor: I think the citation is missing for the FISTA algorithm in the methods.

We thank the reviewer for their comment. The citation was present on page 22 of the methods but was not obviously linked to the algorithm; we have moved the citation forward in the sentence to make it clearer that the method is associated with the cited paper.

REVIEWER COMMENTS

Reviewer #1 (Remarks to the Author):

The authors have responded to all my comments. I find this article very interesting,

Reviewer #3 (Remarks to the Author):

The author's response addresses some of my comments, and has clarified certain misunderstandings in my reading of the paper.

The question of significance remains a matter of judgement. The revised manuscript places less weight on the possible relevance to the fovea, and this is a helpful step.

I would like to clarify, following the author's response, why the image reconstruction problem is much easier for small motion than for large motion. One reason is that in the former case it is possible to obtain a fairly accurate, high resolution estimate of the image even with motion completely ignored; and having a good high resolution estimate of the image greatly simplifies the inference of the trajectory, which then enables refinement of the estimate. The essence of the difficulty in foveal vision is that there is no simple way to reconstruct the image without a reconstruction of the trajectory, and at the same time, trajectory reconstruction is difficult without knowledge of the image content. These insights have been discussed in previous works. The differences relate also to implementational aspects of an algorithm that could solve the problem. For motion that spans less than one ganglion cell, adjusting for motion only requires interaction with adjacent ganglion cells and is therefore a local computation, whereas an algorithmic solution of the problem for the fovea requires dynamic rerouting of information across multiple cells.

Furthermore, as noted by the other reviewers, research on the perceptual consequences of fixational drift has focused on the fovea or parafovea, whereas this question was not central in previous research on peripheral vision. I do agree with reviewer 2 that the result is still interesting, especially because the benefits of the motion are not realized with the Zero-LNBRC-dCNN reconstruction, which ignores motion. This suggests that even in the periphery the brain should take motion into account to achieve optimal reconstruction, yet it is unknown whether the brain actually performs such a computation.

I still find the lack of interpretability on why motion is beneficial dissatisfying, especially when considering that the reconstruction algorithm was selected for interpretability. Previous works came to the conclusion that small amplitude motion (as in the present study) should improve reconstruction due to the enhanced activation of RGCs, yet it is not clear whether this is correct, and alternative proposals were put forth for potential benefits of the motion. The instability of the generative model, mentioned in response to my previous point 3 is a bit worrying; even with this limitation, there may be ways to address the question raised in point 2 other than in simulations, such as dilution of spikes, or tuning down the weight of evidence coming from the spikes in the inference algorithm, to mimic the effect of a small change in the firing rates. This question is quite central to the main conclusion of the work, more so than other questions that are addressed in the manuscript using manipulations of the reconstruction algorithm.

Additional comments:

The text below the scale bar in figure 2 is unclear, at least in the rendering of the figure on my pdf viewer. Wide horizontal spaces are interspersed between the digits, the micron symbol, and the 'm' symbol.

Finally, a new paragraph in the Discussion acknowledges the differences in scale between the peripheral region examined in this work, and the fovea. This is an important addition to the discussion. The numbers provided in the paragraph should be revised, since they depict a quantitatively incorrect picture of these differences, which are in fact much more dramatic than suggested. The present choice of words ("several receptive fields" in the fovea vs. "1-2 RGC receptive fields" in the experiment) creates the impression that we are dealing with a factor of 5 or perhaps even less, between receptive field distances in the peripheral region, compared to the fovea. However, the actual factor is of order 50, since the density of photoreceptors in the center of the fovea is about 300,000 per square mm, with dedicated RGCs for each photoreceptor, and the RGC density in the experiment is of order 120 per square mm, a factor of order 2500 in the density per unit area. The text is generating an incorrect impression of these differences because it describes motion of 1-2 RF distances in the experiment, whereas the root mean square root distance over 500ms is about 35 microns, only a fraction of the average distance between adjacent midget cells (and an even tinier fraction for the parasol cells); and it downplays the extent of motion on the foveal scale, where the root mean square motion over 500 ms corresponds to more than 10 RF distances. It is important to convey correctly that the difference in the relative extent of motion is huge, almost two orders of magnitude.

Reviewer #4 (Remarks to the Author):

I believe the authors have more than adequately addressed my comments and those of the other reviewers. I have no further comments.

Reviewer #1 (Remarks to the Author):

The authors have responded to all my comments. I find this article very interesting,

Thank you.

Reviewer #4 (Remarks to the Author):

I believe the authors have more than adequately addressed my comments and those of the other reviewers. I have no further comments.

Thank you.

Reviewer #3 (Remarks to the Author):

The author's response addresses some of my comments, and has clarified certain misunderstandings in my reading of the paper.

Thank you for raising these important and interesting questions. We agree that addressing them has significantly improved the manuscript (as do the other reviewers).

The question of significance remains a matter of judgement. The revised manuscript places less weight on the possible relevance to the fovea, and this is a helpful step.

I would like to clarify, following the author's response, why the image reconstruction problem is much easier for small motion than for large motion. One reason is that in the former case it is possible to obtain a fairly accurate, high resolution estimate of the image even with motion completely ignored; and having a good high resolution estimate of the image greatly simplifies the inference of the trajectory, which then enables refinement of the estimate. The essence of the difficulty in foveal vision is that there is no simple way to reconstruct the image without a reconstruction of the trajectory, and at the same time, trajectory reconstruction is difficult without knowledge of the image content. These insights have been discussed in previous works. The differences relate also to implementational aspects of an algorithm that could solve the problem. For motion that spans less than one ganglion cell, adjusting for motion only requires interaction with adjacent ganglion cells and is therefore a local computation, whereas an algorithmic solution of the problem for the fovea requires dynamic rerouting of information across multiple cells.

These are interesting points about the relative degree of complexity between foveal vs. peripheral RGC decoding. Our paper is, of course, about the latter. We have inserted a phrase in Discussion to highlight the point about many cells for readers who may be wondering about the same issues. We have kept this paragraph qualitative like the rest of the Discussion.

In the spirit of ongoing discussion of this point, our intuition (shared by the other reviewers, who found our responses on this issue sufficient) is that increasing the magnitude of eye drift relative to receptive field size will more strongly modulate RGC firing, and could make foveal spike trains more informative. Furthermore, because the motion associated with drift eye movements is shared across the entire retina, compensation for drift could in principle use information from RGCs throughout the retina, including the fovea. Thus our estimate of the information about drift eye movements available in RGC spike trains is strictly a lower bound. Having said that, we agree with the reviewer that we don't actually know how the foveal reconstruction problem will work until we can perform those experiments. We continue to advance our recording technology, but this will of course take time. In the meantime, we consider the present results to be a significant step toward answering these questions and we hope the reviewer agrees.

Furthermore, as noted by the other reviewers, research on the perceptual consequences of fixational drift has focused on the fovea or parafovea, whereas this question was not central in previous research on peripheral vision. I do agree with reviewer 2 that the result is still interesting, especially because the benefits of the motion are not realized with the Zero-LNBRC-dCNN reconstruction, which ignores motion. This suggests that even in the periphery the brain should take motion into account to achieve optimal reconstruction, yet it is unknown whether the brain actually performs such a computation.

We agree.

I still find the lack of interpretability on why motion is beneficial dissatisfying, especially when considering that the reconstruction algorithm was selected for interpretability. Previous works came to the conclusion that small amplitude motion (as in the present study) should improve reconstruction due to the enhanced activation of RGCs, yet it is not clear whether this is correct, and alternative proposals were put forth for potential benefits of the motion. The instability of the generative model, mentioned in response to my previous point 3 is a bit worrying; even with this limitation, there may be ways to address the question raised in point 2 other than in simulations, such as dilution of spikes, or tuning down the weight of evidence coming from the spikes in the inference algorithm, to mimic the effect of a small change in the firing rates. This question is quite central to the main conclusion of the work, more so than other questions that are addressed in the manuscript using manipulations of the reconstruction algorithm.

We agree with the reviewer that improvement in reconstruction due to eye drift could be associated with other effects beyond the firing rate, such as altered correlation structure or spatial sampling. We did indeed make a serious attempt to explore this further because we also found it to be an interesting question. Unfortunately, we found that convincingly isolating these effects in real data is extraordinarily difficult. For example, artificial perturbation of spike trains to manipulate the firing rate also disrupts other important aspects, including the stimulus response relationship, temporal structure, and correlation structure between cells. In the end we were not able to produce a convincing analysis of this issue. Future steps could include additional experiments, or perhaps an extensive analysis based on detailed simulations, designed to pick apart these issues.

Additional comments:

The text below the scale bar in figure 2 is unclear, at least in the rendering of the figure on my pdf viewer. Wide horizontal spaces are interspersed between the digits, the micron symbol, and the 'm' symbol.

We apologize for the artifact in the document produced during PDF conversion, and will correct the error in the final manuscript.

Finally, a new paragraph in the Discussion acknowledges the differences in scale between the peripheral region examined in this work, and the fovea. This is an important addition to the discussion. The numbers provided in the paragraph should be revised, since they depict a quantitatively incorrect picture of these differences, which are in fact much more dramatic than suggested. The present choice of words (“several receptive fields” in the fovea vs. “1-2 RGC receptive fields” in the experiment) creates the impression that we are dealing with a factor of 5 or perhaps even less, between receptive field distances in the peripheral region, compared to the fovea. However, the actual factor is of order 50, since the density of photoreceptors in the center of the fovea is about 300,000 per square mm, with dedicated RGCs for each photoreceptor, and the RGC density in the experiment is of order 120 per square mm, a factor of order 2500 in the density per unit area. The text is generating an incorrect impression of these differences because it describes motion of 1-2 RF distances in the experiment, whereas the root mean square root distance over 500ms is about 35 microns, only a fraction of the average distance between adjacent midget cells (and an even tinier fraction for the parasol cells); and it downplays the extent of motion on the foveal scale, where the root mean square motion over 500 ms corresponds to more than 10 RF distances. It is important to convey correctly that the difference in the relative extent of motion is huge, almost two orders of magnitude.

We certainly would not want to misrepresent the substantial differences between fovea and periphery. We have revised the language in this paragraph to emphasize the large differences in receptive field size and density. We have kept it qualitative, like the rest of the Discussion. Most importantly, of course, we provide the actual numbers in Results so that readers can compare objectively and quantitatively (please see the paragraph

that begins: “Interestingly, the enhancement occurred even though the fixational drift was small...”). We hope this provides the reader with sufficient information to consider the problem, as the reviewer did. However, if the reviewer would like to see additional metrics beyond what we have provided in this paragraph, we would be happy to address any specific suggestions that would improve the clarity of the manuscript. We thank the reviewer for their attention to detail on this issue and hope that in the future we can address these questions by further advancing our recording technology.

REVIEWERS' COMMENTS

Reviewer #3 (Remarks to the Author):

Thanks for the detailed response and the revision of the Discussion. I'd prefer to see a quantitative statement not only about the density of receptive fields in the recording area but also in the fovea, because the differences are truly striking and it is important, for future research, not to create an impression that the questions on foveal vision are resolved. I don't insist on this, however. I found the current phrasing in the Discussion to be sufficiently precise. Given the enthusiasm of the other reviewers the manuscript should clearly be accepted for publication and I do not wish to delay the process further. I'd like to thank the authors for their constructive responses, and I'm very much looking forward to future research, building on potential advancements of the technology, that may enable similar analysis in the fovea.

REVIEWERS' COMMENTS

Reviewer #3 (Remarks to the Author):

Thanks for the detailed response and the revision of the Discussion. I'd prefer to see a quantitative statement not only about the density of receptive fields in the recording area but also in the fovea, because the differences are truly striking and it is important, for future research, not to create an impression that the questions on foveal vision are resolved. I don't insist on this, however. I found the current phrasing in the Discussion to be sufficiently precise. Given the enthusiasm of the other reviewers the manuscript should clearly be accepted for publication and I do not wish to delay the process further. I'd like to thank the authors for their constructive responses, and I'm very much looking forward to future research, building on potential advancements of the technology, that may enable similar analysis in the fovea.

We thank the reviewer for their comments. In the final version of the manuscript, to further emphasize possible differences between fovea and periphery, we have added an approximate quantitative comparison of the number of cones traversed in the fovea vs. the periphery to the Discussion.